EMBO
Molecular Medicine

# Deletion of the von Hippel–Lindau gene causes sympathoadrenal cell death and impairs chemoreceptor-mediated adaptation to hypoxia

David Macías[1,2,3], Mary Carmen Fernández-Agüera[1,2,3], Victoria Bonilla-Henao[1,2,3] & José López-Barneo[1,2,3,*]

## Abstract

Mutations of the von Hippel–Lindau (*VHL*) gene are associated with pheochromocytomas and paragangliomas, but the role of VHL in sympathoadrenal homeostasis is unknown. We generated mice lacking Vhl in catecholaminergic cells. They exhibited atrophy of the carotid body (CB), adrenal medulla, and sympathetic ganglia. *Vhl*-null animals had an increased number of adult CB stem cells, although the survival of newly generated neuron-like glomus cells was severely compromised. The effects of Vhl deficiency were neither prevented by pharmacological inhibition of prolyl hydroxylases or selective genetic down-regulation of prolyl hydroxylase-3, nor phenocopied by hypoxia inducible factor overexpression. Vhl-deficient animals appeared normal in normoxia but survived for only a few days in hypoxia, presenting with pronounced erythrocytosis, pulmonary edema, and right cardiac hypertrophy. Therefore, in the normal sympathoadrenal setting, *Vhl* deletion does not give rise to tumors but impairs development and plasticity of the peripheral $O_2$-sensing system required for survival in hypoxic conditions.

**Keywords** adult carotid body neurogenesis; intolerance to hypoxia; sympathoadrenal tumorigenesis; Vhl-deficient mouse model; von Hippel–Lindau protein

**Subject Categories** Cancer; Development & Differentiation; Stem Cells

## Introduction

During embryogenesis, most neural crest-derived sympathetic precursor cells undergo c-jun-dependent apoptosis as the availability of trophic factors (particularly nerve growth factor) becomes limiting (Estus *et al*, 1994). In recent years, *in vitro* experiments on PC12 cells have suggested that the von Hippel–Lindau (VHL) protein might participate in the molecular cascade leading to apoptosis of sympathetic progenitor cells and that impairment of this protein could predispose to pheochromocytomas, a tumor of the adrenal gland, in adulthood (Lee *et al*, 2005). Besides pheochromocytomas, carotid body (CB) paragangliomas and other tumors of neural crest lineage are frequently associated with VHL disease, a hereditary syndrome caused by mutations in the *VHL* gene and characterized by the occurrence of tumors in multiple tissues (Haase, 2005; Kaelin, 2008; Boedeker *et al*, 2009). The best-known function of VHL is to act as the substrate recognition unit of an ubiquitin ligase complex that targets HIFα-subunits for proteasomal degradation (Maxwell *et al*, 1999). However, the role of VHL in the development and homeostasis of the sympathoadrenal system is, as yet, poorly studied. Whereas loss of VHL protein can induce tumors in several organs, it also negatively affects cell survival and proliferation in other tissues (Haase, 2005; Young *et al*, 2008; Li & Kim, 2011). Therefore, inactivation of *VHL* could have differing effects in cells of diverse embryological origin or developmental stage.

To further elucidate the actions of VHL protein, we have generated conditional *Vhl* knockout (KO) mouse models (TH-VHL[KO] and TH-CRE[ER]-VHL[KO] mice) restricted to catecholaminergic (tyrosine hydroxylase—TH—positive) cells to investigate the role played by Vhl in sympathoadrenal development as well as in maintenance of catecholaminergic cells and CB neurogenesis in adulthood. The CB and adrenal medulla (AM) are part of a homeostatic acute $O_2$-sensing system that is essential for survival upon exposure to hypoxia (Weir *et al*, 2005). The CB contains neuron-like, $O_2$-sensitive, glomus (type I) cells that acutely respond to hypoxia by the release of neurotransmitters that stimulate afferent nerve fibers, which activate the brain stem respiratory center and increase sympathetic tone (Lopez-Barneo *et al*, 2001). This organ acts in concert with chromaffin cells of the AM. Activation of the chemosensitive CB-AM axis leads to adaptive hyperventilation and increased cardiac output with redistribution of blood flow to the most $O_2$-demanding organs, such as the brain or the heart. During protracted exposure to low $PO_2$, acclimatization to hypoxia depends on CB hypertrophy and the resulting enhancement of the respiratory drive necessary for

1   Instituto de Biomedicina de Sevilla (IBiS), Hospital Universitario Virgen del Rocío/CSIC/Universidad de Sevilla, Sevilla, Spain
2   Departamento de Fisiología Médica y Biofísica, Facultad de Medicina, Universidad de Sevilla, Sevilla, Spain
3   Centro de Investigación Biomédica en Red sobre Enfermedades Neurodegenerativas (CIBERNED), Madrid, Spain
    *Corresponding author. Tel: +34 955 923007; E-mail: lbarneo@us.es

sustained hyperventilation (Powell et al, 1998; Joseph & Pequignot, 2009). This remarkable proliferative response, uncommon for an organ of neural origin, is achieved thanks to the presence in the adult CB of a resident population of multipotent, neural crest-derived stem cells, which are the glia-like sustentacular (type II) cells (Pardal et al, 2007). Alterations of peripheral neural organs of the sympathoadrenal lineage might lead to dysfunction of the homeostatic acute $O_2$-sensing system with relevant medical impact. Indeed, developmental defects in the CB have been associated with respiratory pathologies in children, such as the sudden infant death or congenital hypoventilation syndromes (see for reviews López-Barneo et al, 2008; Perez & Keens, 2013; Porzionato et al, 2013). Constitutive intolerance to low $PO_2$ could also explain why some individuals are unable to acclimatize to hypoxia and develop complications such as pulmonary hypertension, right heart failure, or brain dysfunction (see Ghofrani et al, 2004; Schou et al, 2012).

Herein, we show that, contrary to generalized beliefs ascribing to VHL a role as tumor suppressor gene (see for references Lee et al, 2005; López-Jiménez et al, 2010; Li & Kim, 2011), Vhl inactivation in rodent catecholaminergic cells in vivo does not lead to tumorigenesis but rather to a marked atrophy of the CB, AM, and sympathetic ganglia. Hypoxia-induced adult CB neurogenesis is also markedly inhibited in mice with the ablation of Vhl alleles. Vhl-deficient animals live normally under normoxic conditions, but show a striking intolerance to systemic hypoxia leading to impending death.

# Results

## Selective disruption of the sympathoadrenal system in Vhl-deficient mice

TH-VHL[KO] mice, with embryonic ablation of Vhl alleles (see Materials and Methods), were viable, fertile, and appeared healthy, reaching adulthood without obvious abnormalities (Supplementary Fig S1A and B). CB and AM dissected from adult TH-VHL[KO] mice did not present any signs of tumor formation. On the contrary, histological analyses showed atrophy of the CB, superior cervical ganglion (SCG), and AM, with a striking decrease in TH[+] cell number. These differences between control and mutant animals were already observed in neonates, but became more apparent during postnatal development (Fig 1A–C; Supplementary Fig S2A and B). CB cells of TH-VHL[KO] mice appeared intermingled with SCG neurons and lacked the cluster-like organization (glomeruli) characteristic of this structure. Quantification of the volume of the CB-SCG TH[+] area clearly showed a marked decrease in size with respect to normal animals (Fig 1B). As a consequence of cell death, the AM almost disappeared by 2–3 months of age (Fig 1C). Abdominal sympathetic ganglia were also affected in TH-VHL[KO] mice (Supplementary Fig S2C). In accord with these structural modifications, the plasma levels of noradrenaline, and particularly adrenaline, were drastically decreased (Fig 1D). Electron microscope analyses demonstrated profound ultrastructural alterations in CB glomus cells of TH-VHL[KO] mice, which showed large vacuoles resembling aberrant autolysosomes, disappearance of the typical dense-core secretory vesicles, and disorganization of nuclear chromatin (Fig 1E). Similar alterations were observed in

adrenal chromaffin cells (Fig 1F). Non-catecholaminergic neural crest-derived cells (such as those in the enteric nervous system or dorsal root ganglion—DRG—neurons) were unaffected by TH promoter-directed Vhl deletion (Supplementary Fig S3A and B). Interestingly, dopaminergic and noradrenergic neurons in the ventral mesencephalon and locus coeruleus, respectively, appeared preserved in juvenile mutant animals (Supplementary Fig S4A–C). These observations suggest that Vhl inactivation in TH[+] cells selectively impairs the development of sympathoadrenal organs, particularly the CB, AM, and sympathetic ganglia, in a cell-autonomous manner.

Since VHL disease-associated tumorigenesis is triggered when the second VHL allele is lost in adult life, we also tested the effects of catecholaminergic-specific Vhl deletion in adult Vhl[Flox/−] mice (TH-CRE[ER]-VHL[KO] mice). These animals, studied 6 months after deletion of the floxed Vhl allele, did not show tumor formation in the CB or AM but a trend to decreased density of TH[+] cells and disorganization of the TH[+] cell clusters (Fig 2A–G). Although macroscopically the CB or AM volumes remained unaltered, impairment of CB function was already detectable in TH-CRE[ER]-VHL[KO] mice (see Fig 8B below). These experiments support the notion that homozygous Vhl loss does not induce tumorigenesis in mouse sympathoadrenal cells.

## Carotid body neurogenesis from adult stem cells is impaired by Vhl deficiency

We tested whether, in addition to its effects on embryonic development, Vhl influenced differentiation and/or survival of newly generated adult CB glomus (type I) cells. It is known that the adult CB is a neurogenic niche containing GFAP[+], glia-like stem cells, that upon exposure to low $O_2$ generate nestin[+] intermediate progenitors which proliferate and, eventually, differentiate into new TH[+] glomus cells and other cell types (Pardal et al, 2007; Platero-Luengo et al, 2014). The number of GFAP[+] CB progenitors was clearly larger in Vhl-ablated animals than in controls and increased progressively with age (Fig 3A). Similarly, the number of proliferating (BrdU[+]) cells in the CB-SCG area was also augmented in TH-VHL[KO] mice with respect to controls (VHL[WT]) (Fig 3B). Ultrastructural studies demonstrated that unlike glomus cells, type II (GFAP[+]) cells, characterized by the form of their nuclei, lack of secretory vesicles, and long processes embracing glomus cells (Platero-Luengo et al, 2014), remained unaffected in TH-VHL[KO] mice (Fig 3C). Therefore, it seems that the loss of differentiated neuron-like glomus cells induced a slow compensatory mechanism that led to an increase in the number of stem cells. In contrast with these findings in the CB, GFAP[+] sustentacular cells in the AM (Suzuki & Kachi, 1995) remained unaffected by the loss of chromaffin cells (Supplementary Fig S5A). Despite the existence of a large population of CB stem cells in the TH-VHL[KO] mice, they did not show the characteristic CB hypertrophy in response to hypoxia (Platero-Luengo et al, 2014) (Supplementary Fig S5B–D). These findings support the notion that damage of Vhl-deficient glomus cells markedly reduced CB responsiveness to lowering $O_2$. Indeed, as shown below, TH-VHL[KO] animals showed a striking intolerance to hypoxia.

To test that GFAP[+] CB stem cells in TH-VHL[KO] mice are actually multipotent and able to differentiate into glomus cells, we

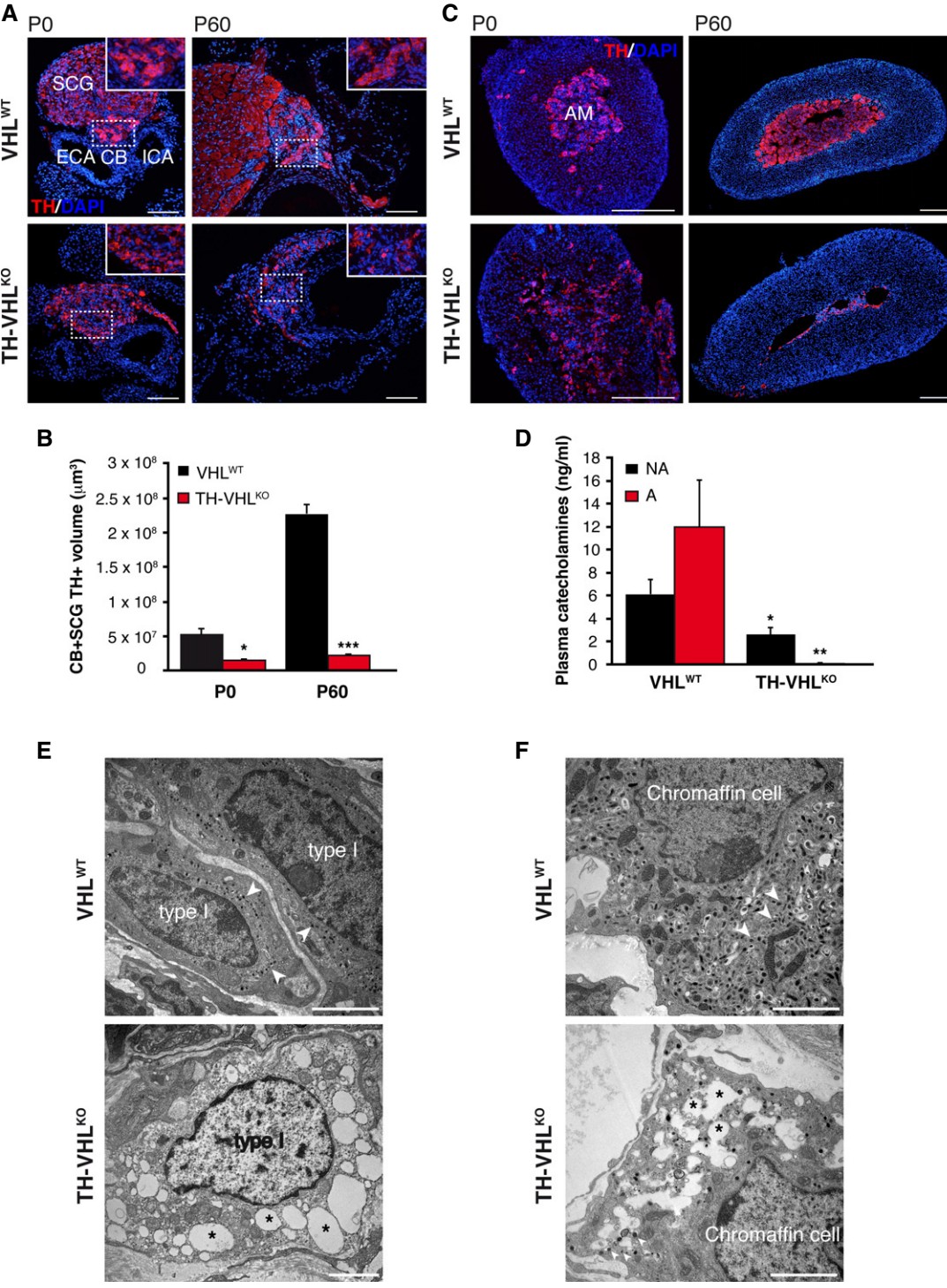

**Figure 1.  Atrophy of sympathoadrenal organs in TH-VHL^KO mice.**

A–D   Immunostaining of TH$^+$ cells in the carotid bifurcation (A) and adrenal gland (C) of TH-VHL$^{KO}$ mice compared with control animals (VHL$^{WT}$). To facilitate comparison, the areas inside the rectangles in (A) are shown in the insets at a higher magnification. ECA, external carotid artery; ICA, internal carotid artery; CB, carotid body; SCG, superior cervical ganglion; AM, adrenal medulla. Scale bars: (A) 100 μm; (C) 200 μm. (B) Quantitative analysis of the CB-SCG TH$^+$ volume in VHL$^{WT}$ and TH-VHL$^{KO}$ mice ($n = 3$ animals per age and genotype). The loss of TH$^+$ cells in mutant mice, already clearly evident at birth (P0), was accentuated during the first postnatal months (P60). *$P = 0.02$, ***$P = 0.000002$ (unpaired two-tailed $t$-test). (D) Plasma catecholamine levels measured by HPLC ($n = 5$ animals 8–12 weeks old per genotype). *$P = 0.04$, **$P = 0.008$ (unpaired two-tailed $t$-test).

E, F   Electron micrograph illustrating ultrastructural alterations observed in glomus (E) and chromaffin (F) cells of TH-VHL$^{KO}$ mice compared with VHL$^{WT}$ animals (2-month-old mice). The presence of numerous dense-core vesicles (arrowheads) and multiple enlarged vacuoles (asterisks) throughout the cytoplasm of Vhl-deficient type I and chromaffin cells are indicated. Scale bars: 2 μm. This figure is accompanied by Supplementary Figs S2, S3 and S4.

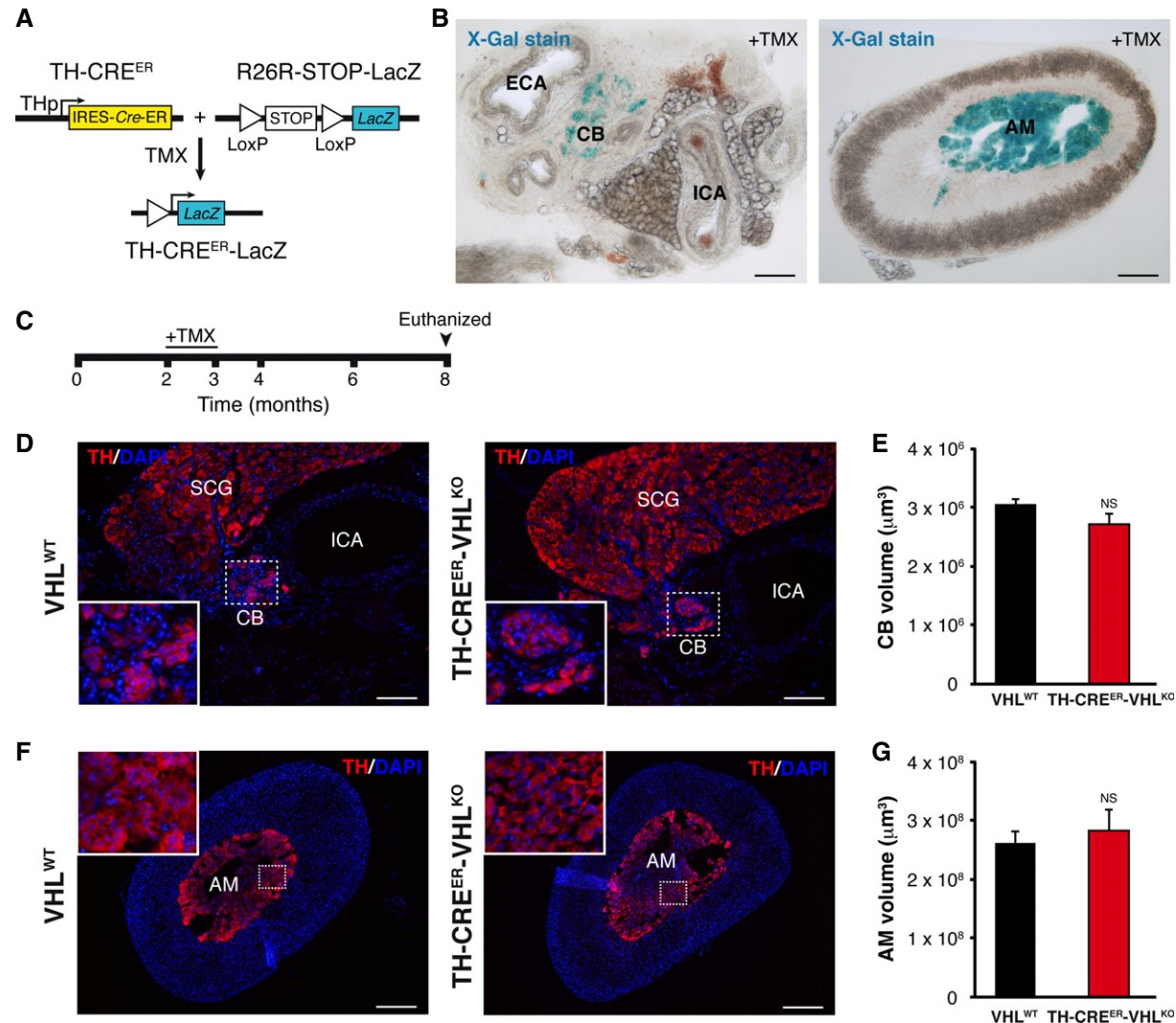

**Figure 2. Catecolaminergic-specific homozygous *Vhl* loss in adult mice does not promote tumor formation.**

A  Scheme of the $Cre^{ER}$-mediated site-specific homologous recombination in adult catecholaminergic cells after tamoxifen administration.

B  X-Gal staining of carotid bifurcation (left) and adrenal gland (right) slices of the TH-$CRE^{ER}$-LacZ mouse to highlight the *LacZ* reporter gene expression after tamoxifen (TMX) treatment.  Scale bars: 100 μm for left panel and 200 μm for right panel.

C  Experimental design for the analysis of the TMX-treated mice.

D  TH immunostaining in the carotid bifurcation of TH-$CRE^{ER}$-$VHL^{KO}$ animals compared with $VHL^{WT}$ mice 5 months after the end of TMX treatment showing no evident changes in glomus cell number although the glomeruli were partially disaggregated. To facilitate comparison, the areas inside the rectangles are shown in the insets at a higher magnification. ICA, internal carotid artery; CB, carotid body; SCG, superior cervical ganglion. Scale bars: 100 μm.

E  Quantification of the CB volume in the TH-$CRE^{ER}$-$VHL^{KO}$ mice compared with control animals studied 5 months after TMX-induced *Vhl* deletion ($n = 4$ per genotype). NS, non-significant (unpaired two-tailed *t*-test).

F  TH immunostaining in the adrenal gland of TH-$CRE^{ER}$-$VHL^{KO}$ animals compared with $VHL^{WT}$ mice after TMX treatment revealing no tumor formation. The insets show the disorganization of clusters of chromaffin cells in the TH-$CRE^{ER}$-$VHL^{KO}$ mice. AM, adrenal medulla. Scale bars: 200 μm.

G  Quantification of the AM volume in the TH-$CRE^{ER}$-$VHL^{KO}$ mice compared with control animals after TMX-induced *Vhl* deletion ($n = 4$ per genotype). NS, non-significant (unpaired two-tailed *t*-test).

performed clonal neurosphere assays (Platero-Luengo *et al*, 2014). CBs of wild-type ($VHL^{WT}$) and TH-$VHL^{KO}$ mice were enzymatically dispersed, and typical CB neurospheres were generated from these cells after 8 days in culture (Fig 4A). However, the number of neurospheres was greater (Fig 4B) and their diameter larger (Fig 4C) in TH-$VHL^{KO}$ than in the $VHL^{WT}$ mice, a result compatible with the increased number of stem cells and cell proliferation in the Vhl-deficient CB (see above and Fig 3) Immunocytochemical

analysis of sectioned neurospheres produced from $VHL^{WT}$ and TH-$VHL^{KO}$ mice revealed the presence of nestin$^+$ cells inside the core of the neurosphere similar to that previously described for rat CB neurospheres. However, small clusters of TH$^+$ cells, which appeared in the wild-type neurospheres after a few days in culture due to differentiation of progenitor cells into glomus cells (Pardal *et al*, 2007), were not observed in preparations from TH-$VHL^{KO}$ mice ($n = 5$ experiments) (Fig 4D). These data indicated

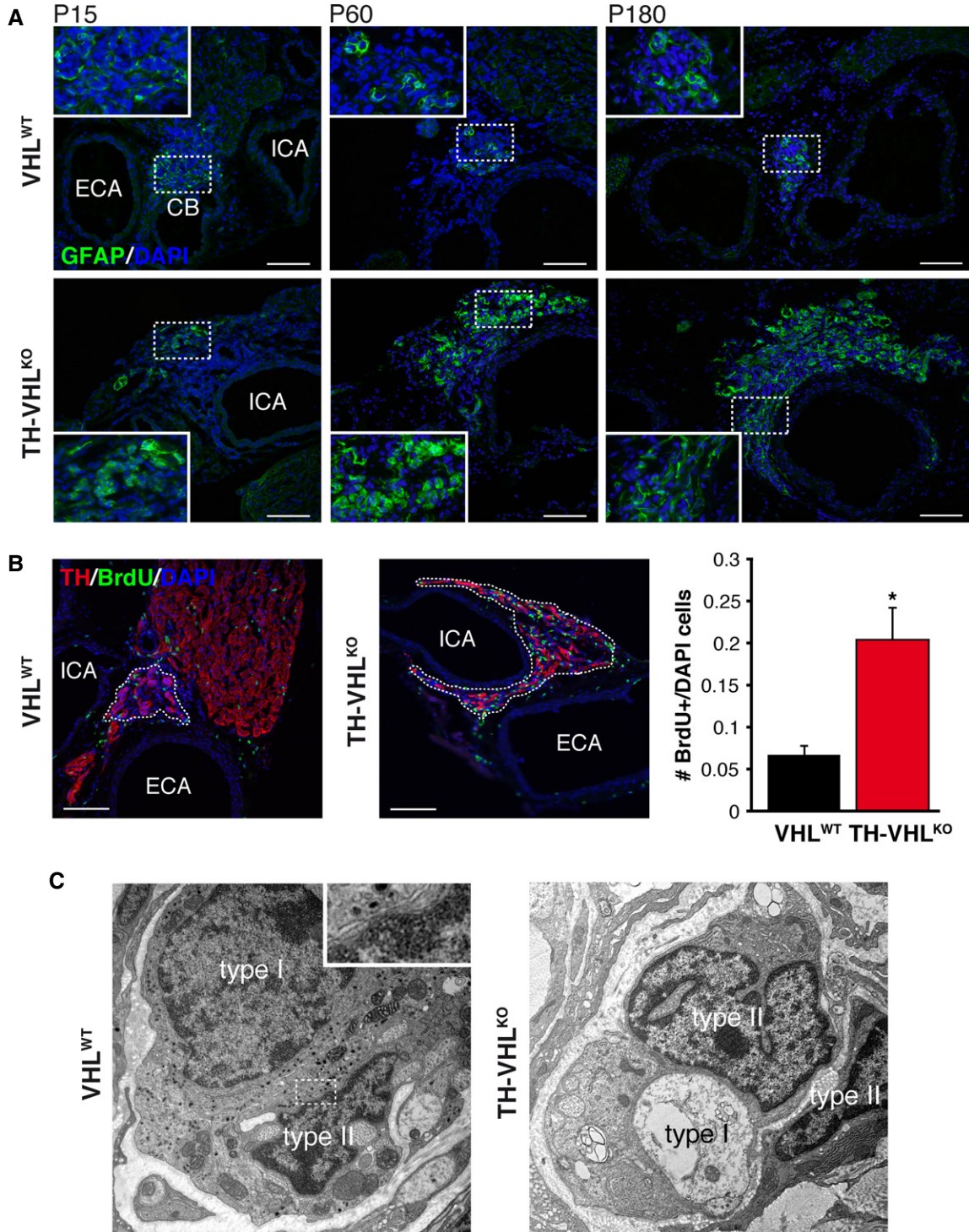

**Figure 3.  Increase of GFAP$^+$ CB stem cell number in TH-VHL$^{KO}$ mice.**

A  GFAP immunostaining illustrating the large increase of GFAP$^+$ cells in the carotid bifurcation of TH-VHL$^{KO}$ animals compared with VHL$^{WT}$ mice (from P15 to P180). The areas inside the rectangles are shown at a higher magnification in the insets. ECA, external carotid artery; ICA, internal carotid artery; CB, carotid body. Scale bars: 100 μm.

B  Left. Incorporation of BrdU to proliferating cells in the carotid bifurcation of VHL$^{WT}$ and TH-VHL$^{KO}$ mice maintained in normoxia. Right. Number of BrdU$^+$ cells versus total cells inside the area occupied by TH$^+$ cells in the CB (VHL$^{WT}$) or the CB-SCG region (TH-VHL$^{KO}$). Scale bars: 100 μm. Cell counting was performed on 4 animals (8 weeks old) per genotype. *$P$ = 0.02 (unpaired two-tailed *t*-test).

C  Electron micrograph showing the normal appearance of CB type II cells in 2-month-old VHL$^{WT}$ and TH-VHL$^{KO}$ mice. Typical synaptic-like contact between type I and type II cell is shown in the inset at higher magnification. Scale bars: 2 μm. This figure is accompanied by Supplementary Fig S5.

that, although progenitor cells in the TH-VHL$^{KO}$-neurospheres are highly proliferative, they seem to be unable to give rise to new TH$^+$ cells, thereby explaining the lack of neurogenesis *in vivo* described above. This supposition was confirmed by culturing neurospheres on adherent substrate to promote differentiation. In contrast to VHL$^{WT}$-derived neurospheres, where mature TH$^+$ cells were present in all cultures ($n = 15$), TH$^+$ cells were not, or very rarely, seen in the neurosphere cultures from TH-VHL$^{KO}$ animals ($n = 15$). In some cultures ($n = 5$), we observed cells in mitosis that expressed both nestin and TH (Fig 4E left-bottom). CB progenitor cells in neurospheres from both VHL$^{WT}$ and TH-VHL$^{KO}$ mice were able to give rise to new smooth muscle actin-positive (SMA$^+$) cells (Fig 4E right), another cell type derived from CB stem cells (Pardal *et al*, 2007). Hence, CB stem cells from TH-VHL$^{KO}$ preserve their multipotency and are able to differentiate into SMA$^+$ and TH$^+$ cells. However, it appears that newly formed glomus cells are unable to differentiate to TH$^+$ cells or die as soon as they express TH, and *Vhl* is inactivated by Cre-mediated site-specific recombination. It would seem therefore that normal Vhl function is thus required not only for proper embryonic development of the sympathoadrenal organs but also for the survival of newly formed adult CB glomus cells.

## Vhl deficiency is not compensated for by ablation of *Phd3* or Hif modulation

Phd3 is an $O_2$-sensing hydroxylase that has pro-apoptotic actions (Lee *et al*, 2005; Bishop *et al*, 2008; reviewed in Schlisio, 2009). We consequently investigated whether the catecholaminergic cell loss due to Vhl deficiency might be as a consequence of Hif-α stabilization and subsequent Phd3 induction. We selected specific shRNA lentiviral vectors (LVs) for *Phd3* silencing (Fig 5A) and then transduced with these vectors dispersed CB cells that were used for neurosphere generation and differentiation assays. In all experiments ($n = 9$ for each condition) with VHL$^{WT}$ neurospheres (with or without LVs), we observed the generation of newly differentiated TH$^+$ cells (Fig 5B top), and the number of these cells slightly increased in response to *Phd3* down-regulation (Fig 5C). As indicated in the preceding section, newly generated TH$^+$ cells were practically absent in the experiments performed with TH-VHL$^{KO}$ neurospheres ($n = 9$) (Fig 5B bottom left). *Phd3* silencing in the TH-VHL$^{KO}$ background favored the generation of some TH$^+$ cells in 5 of 9 experiments (Fig 5B bottom), but this effect was quantitatively negligible as the total number of TH$^+$ cells present was small (Fig 5C). Similar differentiation assays

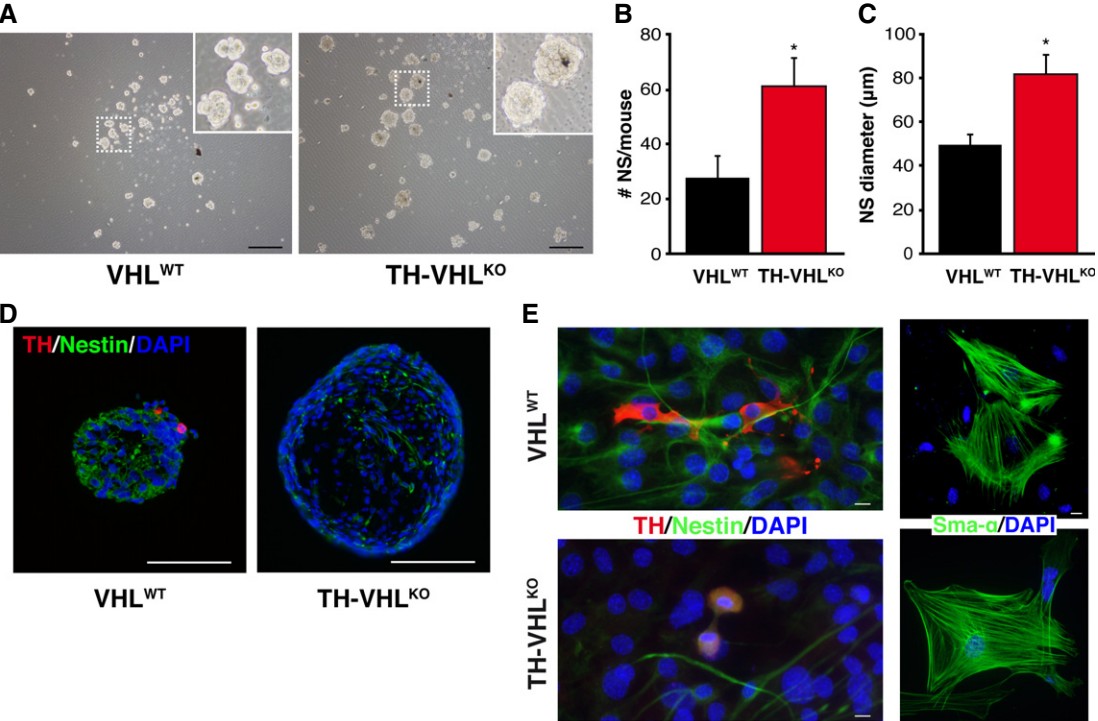

**Figure 4.  Multipotency and differentiation of GFAP$^+$ CB stem cells.**

A    Bright field images of typical CB neurospheres formed from VHL$^{WT}$ and TH-VHL$^{KO}$ mice (8–16 weeks old) after 8 days in culture. Scale bars: 200 μm. The areas inside the rectangles are shown in the insets at higher magnification.

B, C  Number (B) and diameter (C) of CB-derived neurospheres generated from VHL$^{WT}$ and TH-VHL$^{KO}$ animals ($n = 5$ per genotype). *$P = 0.01$ (unpaired two-tailed *t*-test).

D    Sectioned CB neurospheres illustrating the presence of nestin$^+$ cells in VHL$^{WT}$ and TH-VHL$^{KO}$-derived neurospheres and the absence of TH$^+$ cells in TH-VHL$^{KO}$-derived colonies. Scale bars: 100 μm.

E    Immunocytochemical analysis of 8-day-old CB neurospheres cultured for 3 days on adherent substrate. Left panels. Newly differentiated TH$^+$ cells were found only in cultures of VHL$^{WT}$-derived neurospheres ($n = 15$). Some cells undergoing mitosis and expressing both nestin and TH were seen in neurosphere cultures from TH-VHL$^{KO}$ mice ($n = 5$ of 15 experiments). Right panels. Neurosphere cultures from both VHL$^{WT}$ and TH-VHL$^{KO}$ mice gave rise to SMA$^+$ cells. Scale bars: 10 μm.

were performed in neurospheres with variable HIF levels. Neuro-spheres were treated with 0.5 mM dimethyloxalylglycine (DMOG) to inhibit prolyl hydroxylases, thus permitting us to test for the effect of HIF stabilization *in vitro*. In 5 of 6 experiments, incuba-tion of TH-VHL[KO] neurospheres with DMOG favored the appear-ance of some TH[+] cells; however, this effect was also quantitatively very small (Fig 5B and C). Similarly, transduction of cells with lentiviral vectors for *Hif-1α* and *Hif-2α* silencing did not produce a significant increase in the number of newly gener-ated TH[+] cells (Fig 5D). Taken together, these data indicate that the lack of stem cell-dependent glomus cell differentiation, observed in Vhl-deficient animals, is not, or only marginally, compensated for by the Phd3 deficit, generalized prolyl hydroxy-lase inhibition, or *Hif* down-regulation.

We further investigated whether Phd3 can compensate for Vhl protein deficiency in a mouse model with general ablation of *Phd3* and deletion of *Vhl* in catecholaminergic cells. As previously described (Bishop *et al*, 2008), ablation of *Phd3* (VHL[WT];PHD3[KO]) resulted in a slight hypertrophy of the CB and AM (Fig 6A and C, top panels). Ablation of *Vhl* on the *Phd3* knockout background (TH-VHL[KO];PHD3[KO]) resulted in strong reduction of TH[+] paren-chyma in the CB-SCG area (Fig 6A and B) similar to that observed on TH-VHL[KO] animals. TH-VHL[KO];PHD3[KO] mice also showed higher number of GFAP[+] cells in the CB region and marked cell loss in the AM (Fig 6A and C, bottom panels). This is a phenotype qualitatively similar to that described above for TH-VHL[KO] mice (see Figs 1 and 3). CB neurosphere differentiation assays for the TH-VHL[KO];PHD3[KO] mice yielded TH[+] cells in the six experiments performed, thus indicating that *Phd3* ablation does not alter multipotency and that it favors the catecholaminergic differentiation of CB stem cells. However, as occurred in the knockdown experiments (Fig 5), the effect of *Phd3* deletion on the total number of differentiated TH[+] cells, although statistically significant, was quantitatively negligible (Fig 6D and E). Comparison of the number of newly generated TH[+]

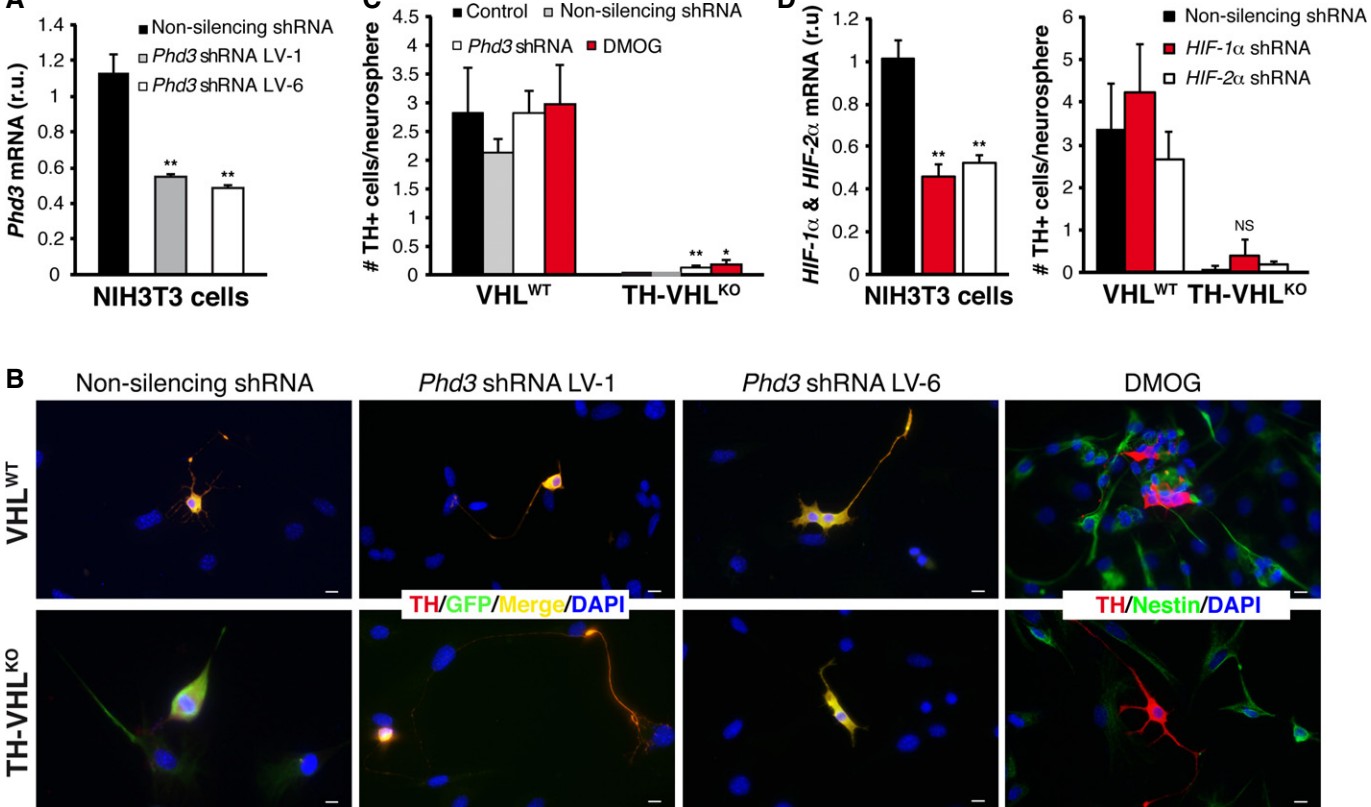

**Figure 5. *Phd3*, *Hif-1α*, and *Hif-2α* down-regulation and prolyl hydroxylase inhibition with DMOG in CB neurosphere cultures.**

A    *Phd3* mRNA levels in NIH3T3 cells 48 h after transduction with non-silencing and specific *Phd3* shRNA (*Phd3* shRNA LV-1 and LV-6) lentiviral vectors (*n* = 3 for each condition). Non-silencing shRNA versus *Phd3* shRNA LV-1, **$P$ = 0.005 and non-silencing shRNA versus *Phd3* shRNA LV-6, **$P$ = 0.004 (unpaired two-tailed *t*-test). r.u., relative units.

B    Immunocytochemical analysis of CB neurospheres from VHL[WT] and TH-VHL[KO] mice transduced with non-silencing and *Phd3* shRNA lentiviral vectors (Phd3 shRNA LV-1 and LV-6) or treated with 0.5 mM DMOG. Transduced cells were identified by GFP expression. Some differentiated TH[+] cells were identified in *Phd3*-silenced and DMOG-treated TH-VHL[KO]-derived neurospheres (*n* = 5 of 9 and *n* = 5 of 6 experiments, respectively). Scale bars: 10 μm.

C    Quantification of TH[+] cells per neurosphere (*n* = 9 for control and non-silencing conditions, *n* = 5 for *Phd3* shRNA transduction and *n* = 6 for DMOG administration). *$P$ = 0.02, **$P$ = 0.007 (unpaired two-tailed *t*-test).

D    Left. *Hif-1α* and *Hif-2α* mRNA levels in NIH3T3 cells 48 h after transduction with non-silencing and specific *Hif-1α* and *Hif-2α* shRNA lentiviral vectors (*n* = 3 for each condition). **$P$ = 0.003 (unpaired two-tailed *t*-test). r.u., relative units. Right. Number of TH[+] cells per CB neurosphere (*n* = 4) transduced with the indicated non-silencing and *Hif-α* shRNA LVs. NS, non-significant (unpaired two-tailed *t*-test).

cells in neurospheres from VHL$^{WT}$;PHD3$^{KO}$ and TH-VHL$^{KO}$;PHD3$^{KO}$ indicates that although *Phd3* ablation favored the appearance of some TH$^+$ cells differentiated from neural progenitors, it did not prevent the progressive TH$^+$-cell loss produced by the absence of *Vhl* (Fig 6F). The results of the *in vitro Phd3* knockdown experiments and the *in vivo* ablation of *Phd3* therefore indicate that an absence of this hydroxylase does not prevent cell death in VHL-deficient cells. Similar to the *in vitro* data with DMOG-treated cells (see Fig 5), transgenic over-expression of non-degradable forms of HIF-1α and/or HIF-2α (Kim *et al*, 2006) in sympathoadrenal cells (TH-HIF1$^{dPA}$, TH-HIF2$^{dPA}$ mice) did not phenocopy that associated with Vhl deficiency (Fig 7A–H). In contrast, Hif-2α stabilization in TH$^+$ cells resulted in a significant increase in the CB volume and glomus cell number (Fig 7D and E), a finding compatible with that observed in *Phd3* knockout mice (Bishop *et al*, 2008). These data suggest that sympathoadrenal cell loss induced by a deficit in Vhl does not depend on Hif stabilization.

### HVR and acclimatization to chronic hypoxia are severely impaired in TH-VHL$^{KO}$ animals

Despite a marked atrophy of peripheral O$_2$-sensing organs, TH-VHL$^{KO}$ mice exhibited respiratory parameters in normoxic conditions similar to those of controls (see Supplementary Table S1 and Fig 8). Animal responsiveness to acute hypoxia (hypoxic ventilatory response—HVR) was tested by plethysmography. Figure 8A illustrates the changes of respiratory rate during a cycle of normoxia–hypoxia–normoxia in VHL$^{WT}$ and TH-VHL$^{KO}$ mice. After application of hypoxia (10% O$_2$), VHL$^{WT}$ animals increased their respiration up to a plateau level, which then decreased to basal levels upon returning to normoxia (21% O$_2$) (Fig 8A, black line). In contrast, TH-VHL$^{KO}$ mice, which exhibited a normal respiratory rate in normoxia, failed to hyperventilate in response to hypoxia (Fig 8A, gray discontinuous line). In about half of the trials, TH-VHL$^{KO}$ mice exposed to hypoxia also showed a transient loss of consciousness and marked respiratory depression (Fig 8A, red line). A summary of average respiratory rates during exposure to hypoxia of the various animal models studied is given in Fig 8B. As expected, VHL$^{WT}$ and VHL$^{WT}$;PHD3$^{KO}$ animals showed an increased average respiratory rate in response to hypoxia. In contrast, TH-VHL$^{KO}$ and TH-VHL$^{KO}$;PHD3$^{KO}$ mice failed to show any sign of HVR. TH-CRE$^{ER}$-VHL$^{KO}$ mice studied 6 months after tamoxifen treatment (see Fig 2) also showed partial inhibition of the HVR (Fig 8B). Additional respiratory parameters altered by *Vhl* deletion are presented in Supplementary Table S1. We also measured arterial hemoglobin saturation during acute exposures to mild (14% O$_2$) and more severe (10% O$_2$) hypoxia as an indication of the efficacy of compensatory hyperventilation. Whereas VHL$^{WT}$ and VHL$^{WT}$;PHD3$^{KO}$ mice were able to maintain hemoglobin saturation at 60–65% during exposure to hypoxia, this value dropped to < 40% in TH-VHL$^{KO}$ and TH-VHL$^{KO}$;PHD3$^{KO}$ animals (Fig 8C).

The lack of responsiveness to acute lowering of O$_2$ tension made *Vhl*-deficient animals intolerant to sustained hypoxia. Consistent with plethysmography and hemoglobin saturation recordings, TH-VHL$^{KO}$ mice showed signs of respiratory distress and loss of consciousness after 30–40 s in response to chronic hypoxia conditions; most of them, however, recovered in 2–3 min, although their long-term survival was severely compromised. While VHL$^{WT}$ mice adapted well to chronic hypoxia (10 or 14% O$_2$), none of the

TH-VHL$^{KO}$ or TH-VHL$^{KO}$;PHD3$^{KO}$ mice (in which *Vhl* alleles had been deleted only in catecholaminergic tissues) survived for more than 11 days in either a 10% (Fig 8D) or 14% O$_2$ environment. When maintained under hypoxic conditions, TH-VHL$^{KO}$ mice showed abnormally high hematocrit and EPO plasma levels on day 7 compared with wild-type mice (Fig 8E and F). These animals, with catecholaminergic cell *Vhl* deficiency and lack of the hypoxic hyperventilatory response, also exhibited gross anatomical alterations of the heart, lung, and spleen (Fig 9A–F). The most salient pathological feature was a marked increase in heart size due to enlargement of the right ventricle (Fig 9A,B and D), probably due to pulmonary hypertension. We also found evidence of pulmonary edema and parenchyma micro-hemorrhage in the lungs of TH-VHL$^{KO}$ animals (Fig 9E). Spleens also showed a marked increase in size (Fig 9A and C) and exhibited changes in histological architecture characteristic of extramedullary hematopoiesis (Fig 9F). None of these histological alterations were seen in TH-VHL$^{KO}$ mice living under normoxic conditions (Supplementary Fig S6A–C).

## Discussion

### Role of Vhl in sympathoadrenal development, tumorigenesis, and adult CB neurogenesis

*Vhl* is commonly recognized as a tumor suppressor gene given that its homozygous deletion is known to induce tumors, particularly hemangioblastomas and renal cysts cancers, in a HIF-dependent fashion (see Haase, 2005; Kaelin, 2007 for reviews). A point mutation (R200W) in *Vhl* that is not associated with tumor development recapitulates in mice Chuvash polycythemia via Hif-2α signaling (Ang *et al*, 2002; Hickey *et al*, 2007). These effects are consistent with the most-studied function of VHL as a substrate recognition unit of an ubiquitin ligase complex that targets HIFα for proteasomal degradation (Maxwell *et al*, 1999). VHL mutations can also produce pheochromocytomas and paragangliomas, but the role of VHL in the development and homeostasis of the sympathoadrenal system is still not well understood. *In vitro* experiments on PC12 cells have suggested that Vhl participates in the c-jun-dependent cascade that leads to the apoptosis of sympathetic progenitor cells during the late stages of development (Estus *et al*, 1994; Lee *et al*, 2005). In this model, Phd3, a prolyl hydroxylase with a well-established pro-apoptotic role (reviewed in Schlisio, 2009), acts downstream of Vhl to regulate sympathoadrenal progenitor survival. It was thus postulated that alterations of the Vhl-Phd3 system during embryogenesis could predispose affected individuals to pheochromocytomas in adulthood (Lee *et al*, 2005). In contrast to these findings, our *in vivo* and *in vitro* observations suggest that *Vhl* deletion not only results in sympathoadrenal cell loss but also impairs the differentiation and/or survival of stem cell-derived newly generated glomus cells in the adult CB. Moreover, we did not observe any indication of adrenal or CB tumorigenesis after the deletion of a remaining floxed *Vhl* allele, which produced loss of heterozygosity in adulthood (TH-CRE$^{ER}$-VHL$^{KO}$ mice). Embryonic and adult sympathoadrenal cell death observed in Vhl-deficient animals is independent of genetic (Phd3) or pharmacological (DMOG administration) prolyl hydroxylase inhibition and is not mimicked by either Hif-1α or Hif-2α down-regulation or transgenic Hif-1α and Hif-2α activation.

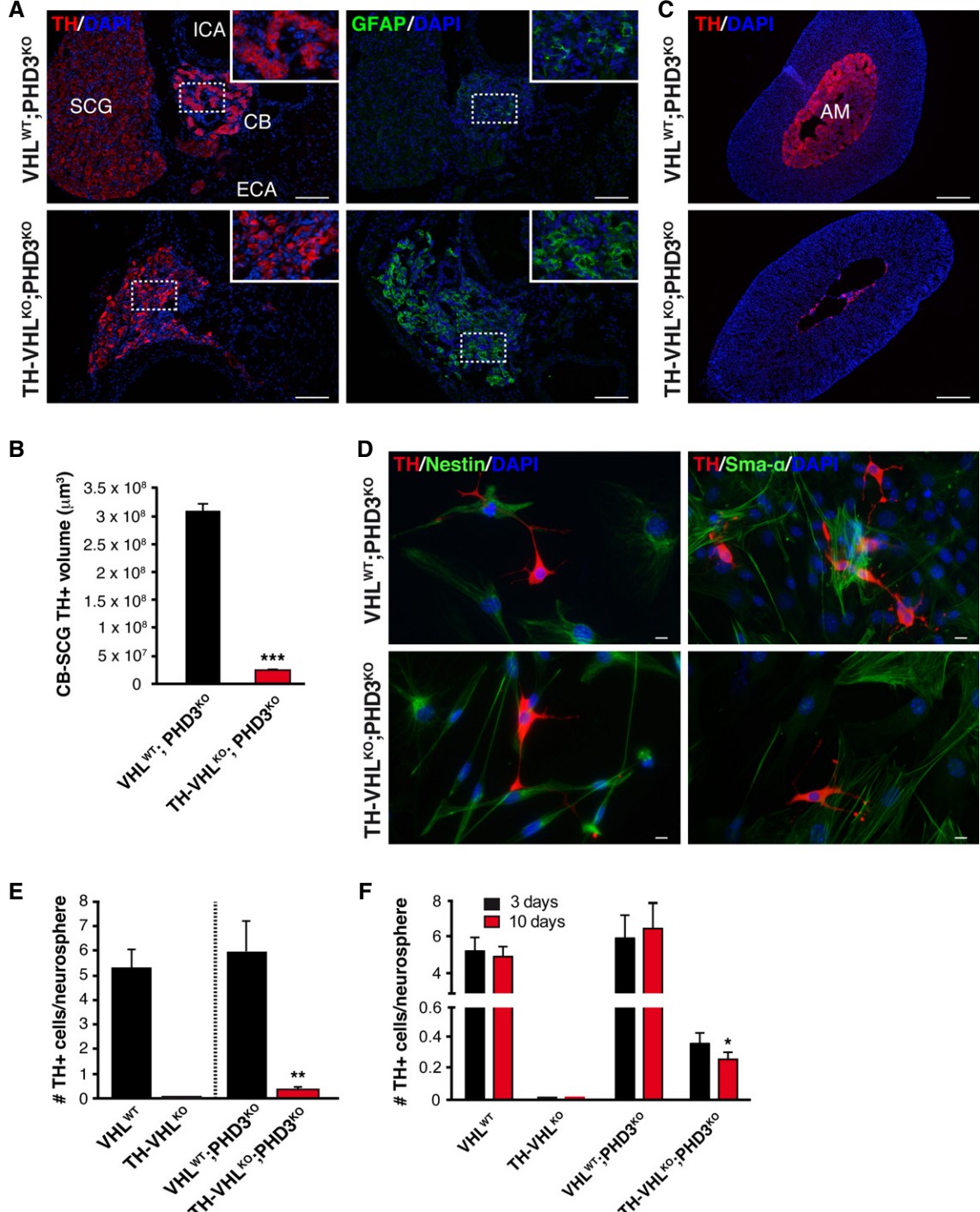

**Figure 6.** *In vivo Phd3* ablation does not prevent the effects of Vhl deficiency.

A   TH and GFAP immunostaining in carotid bifurcation of control (VHL$^{WT}$;PHD3$^{KO}$) and mutant (TH-VHL$^{KO}$;PHD3$^{KO}$) mice at P60. ECA, external carotid artery; ICA, internal carotid artery; CB, carotid body; SCG, superior cervical ganglion. The regions inside the rectangles are shown in the insets at higher magnification. Scale bars: 100 μm.

B   CB-SCG TH$^+$ volume quantification in adult (P60) VHL$^{WT}$;PHD3$^{KO}$ control mice compared with TH-VHL$^{KO}$;PHD3$^{KO}$ mutants (*n* = 3 mice per genotype). ****P* = 0.000002 (unpaired two-tailed *t*-test).

C   Immunofluorescence images of adrenal gland sections stained for TH. AM, adrenal medulla. Scale bars: 200 μm.

D   Representative examples of CB neurosphere cultures from VHL$^{WT}$;PHD3$^{KO}$ and TH-VHL$^{KO}$;PHD3$^{KO}$ mice illustrating the presence of nestin$^+$, TH$^+$ (left panel), and SMA$^+$ (right panel) cells. Scale bars: 10 μm.

E   Number of differentiated TH$^+$ glomus cells per neurosphere generated *in vitro* (*n* = 6 per genotype). ***P* = 0.001 (unpaired two-tailed *t*-test).

F   Quantification of TH$^+$ cells identified after 3 or 10 days in differentiation culture conditions (*n* = 6 per genotype). **P* = 0.04 (unpaired two-tailed *t*-test).

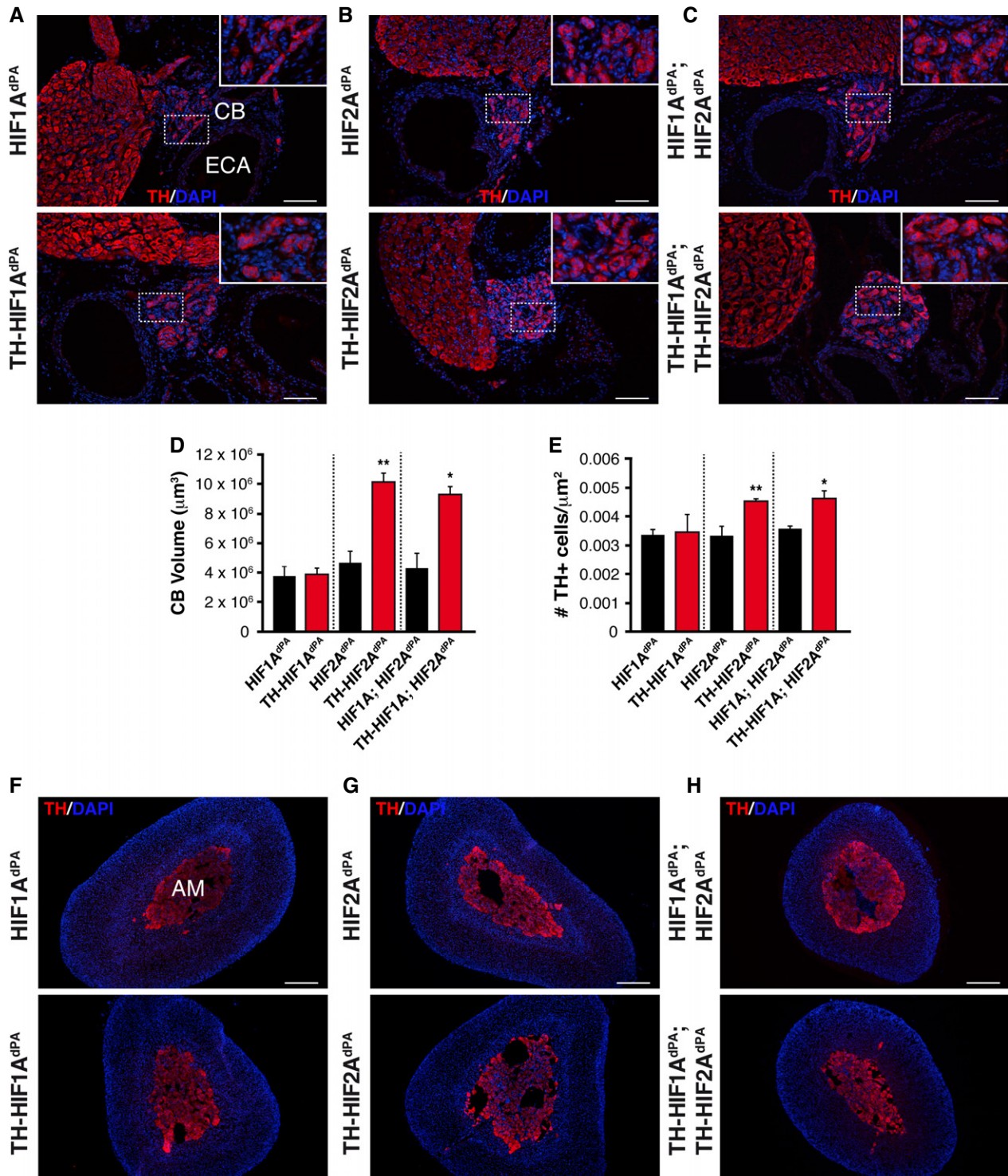

**Figure 7.  Sympathoadrenal effects of HIF-1α and/or HIF-2α over-expression.**

A–C     Immunohistochemical analysis of the carotid body in mice overexpressing non-degradable variants of HIF-1α (A), HIF-2α (B), or both HIF-1α and HIF-2α (C) restricted to the sympathoadrenal system (TH-HIF1A$^{dPA}$, TH-HIF2A$^{dPA}$, and TH-HIF1A$^{dPA}$;TH-HIF2A$^{dPA}$ mouse lines, respectively) compared with controls (HIF1A$^{dPA}$, HIF2A$^{dPA}$, and HIF1A$^{dPA}$;HIF2A$^{dPA}$, respectively). Scale bars: 100 μm. ECA, external carotid artery; CB, carotid body; SCG, superior cervical ganglion.

D, E     Quantification of the carotid body volume (D) and TH$^+$ cell density (E) in the indicated HIF-overexpressing mouse lines compared with controls, respectively. ($n = 3$ per genotype). *$P = 0.03$, **$P = 0.009$ in (D) and *$P = 0.02$, **$P = 0.008$ in (E) (unpaired two-tailed $t$-test).

F–H     Adrenal gland thin sections stained for TH detection illustrating the appearance of chromaffin cells with HIF-1α (F bottom), HIF-2α (G bottom), or both HIF-1α and HIF-2α (H bottom) activation compared with controls (F–H upper panels). Scale bars: 200 μm. All animals were 8–12 weeks old.

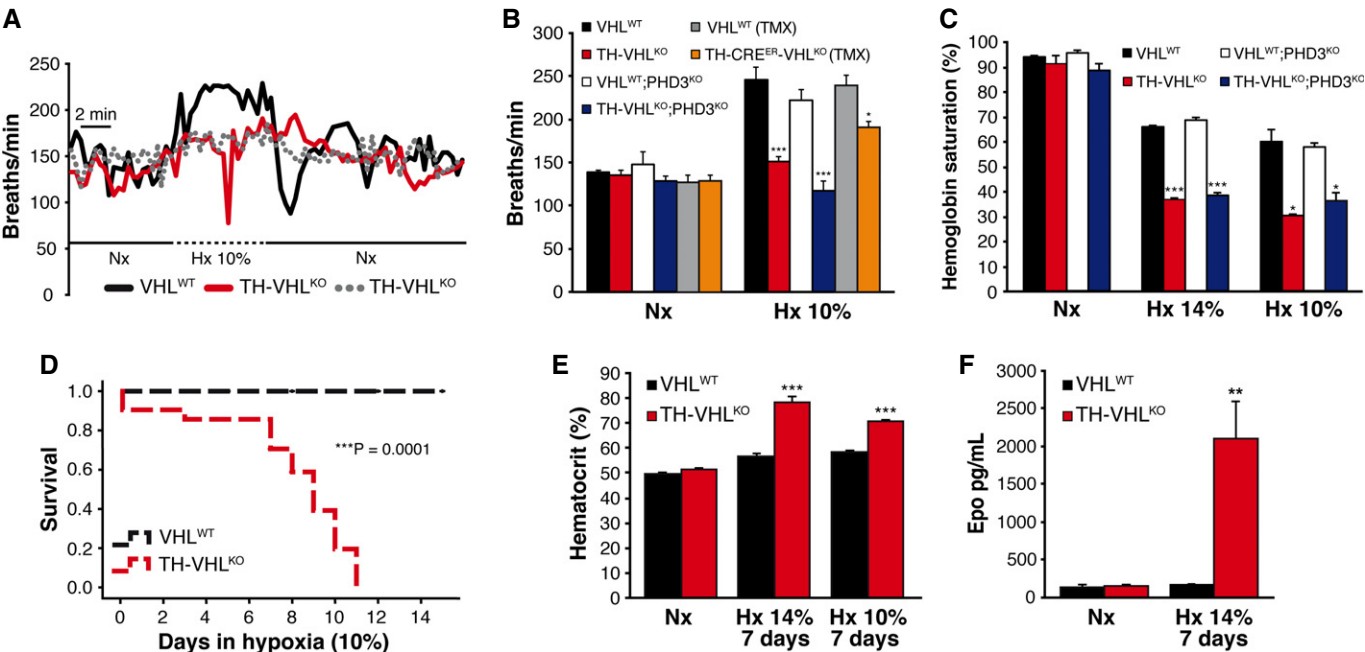

**Figure 8. Hypoxic ventilatory response and acclimatization to chronic hypoxia.**

A  Representative plethysmograph recordings showing changes in respiration rate during a cycle of normoxia–hypoxia–normoxia in VHL$^{WT}$ and TH-VHL$^{KO}$ mice. Nx, normoxia; Hx, hypoxia.

B  Average respiration rate in periods of normoxia and hypoxia for the genotypes studied ($n$ = 6 for VHL$^{WT}$ and TH-VHL$^{KO}$ mice; $n$ = 5 for VHL$^{WT}$;PHD3$^{KO}$ and TH-VHL$^{KO}$; PHD3$^{KO}$ animals; $n$ = 4 for VHL$^{WT}$ and TH-CRE$^{ER}$-VHL$^{KO}$ mice treated with tamoxifen). VHL$^{WT}$ versus TH-VHL$^{KO}$, ***$P$ = 0.0003. VHL$^{WT}$;PHD3$^{KO}$ versus TH-VHL$^{KO}$; PHD3$^{KO}$, ***$P$ = 0.0008. VHL$^{WT}$ versus TH-CRE$^{ER}$-VHL$^{KO}$, *$P$ = 0.03 (unpaired two-tailed $t$-test).

C  Hemoglobin saturation measurement in mice exposed to different levels of O$_2$ (14 and 10% O$_2$; $n$ = 4 per genotype and condition). VHL$^{WT}$ versus TH-VHL$^{KO}$, *$P$ = 0.02, ***$P$ = 0.0007. VHL$^{WT}$;PHD3$^{KO}$ versus TH-VHL$^{KO}$;PHD3$^{KO}$, *$P$ = 0.01, ***$P$ = 0.0005 (unpaired two-tailed $t$-test).

D  Kaplan–Meier survival curve comparing VHL$^{WT}$ and TH-VHL$^{KO}$ mice after exposure to chronic hypoxia ($n$ = 10 per genotype). ***$P$ = 0.0001 (log-rank test).

E  Hematocrit levels after 7 days in chronic hypoxia (14 and 10% O$_2$; $n$ = 6 per genotype at 14% O$_2$ and $n$ = 8 per genotype at 10% O$_2$). VHL$^{WT}$ versus TH-VHL$^{KO}$, ***$P$ = 0.00001 at 14% O$_2$ and ***$P$ = 0.000001 at 10% O$_2$ (unpaired two-tailed $t$-test).

F  EPO plasma levels (determined by ELISA) in mice maintained under normoxic or hypoxic (14% O$_2$) conditions ($n$ = 5 per genotype and condition). **$P$ = 0.003 (unpaired two-tailed $t$-test). Nx, normoxia; Hx, hypoxia. All experiments were performed with 8- to 12-week-old mice.

Therefore, it seems that Vhl is not only necessary for the survival of sympathoadrenal cells during development, but it is also required for maintenance (full differentiation and survival) of these cells in adulthood and for neurogenesis in the adult CB in a manner unrelated to the Vhl-Hif-Phd3 pathway. Interestingly, it has been noted that the bi-allelic loss of *VHL* seems to be incompatible with pheochromocytoma development and that most cases of type 2 VHL disease (high risk of pheochromocytomas) are caused by missense mutations of the *VHL* gene. *Vhl*-null murine embryonic stem (ES) cells generate teratocarcinomas that are smaller than those produced by wild-type ES cells, indicating that the tumor suppressor activity of Vhl is only manifested in a background of other mutations (Mack *et al*, 2003). As previously suggested (see Lee *et al*, 2005 for a detailed discussion), it could be that mutations of the *VHL*-producing pheochromocytoma are gain-of-function mutations that lead to abnormal cell proliferation in the adrenal cell setting. In agreement with these findings, homozygous *Vhl* deletion has been shown to produce impairment of pancreatic beta cell function (Cantley *et al*, 2009) and apoptosis in thymocytes and chondrocytes (Haase, 2005). Although the small size of the CB or AM has precluded any detailed biochemical analysis of sympathoadrenal cell death induced by Vhl deficiency, our electron

microscope studies suggest that it is -compatible with autophagy dysregulation. This idea is in accord with numerous recent reports demonstrating Hif-independent involvement of Vhl in cell senescence, apoptosis, and autophagy (Young *et al*, 2008; Mazure & Pouysségur, 2010; Li & Kim, 2011). Incubation of neurospheres from TH-VHL$^{KO}$ animals with a cocktail including inhibitors of autophagy, apoptosis, and necroptosis significantly increased the generation of viable TH$^+$ cells (data not shown). However, the small effect of this pharmacological treatment precludes any definitive conclusion on the mechanism(s) of cell death in Vhl-deficient catecholaminergic cells. In any instance, our results stress the critical importance of Hif-independent functions of Vhl for sympathoadrenal cell homeostasis.

### Adult CB stem cell population in TH-VHL$^{KO}$ mice

TH-VHL$^{KO}$ mice exhibited a compensatory increase in the number of GFAP$^+$ CB stem cells compared with wild-type mice. This cell population formed multipotent clonal colonies *in vitro,* thus supporting the role of the CB as a neurogenic niche in the peripheral nervous system (Pardal *et al*, 2007). However, as indicated above, differentiation and/or survival of newly generated glomus cells from

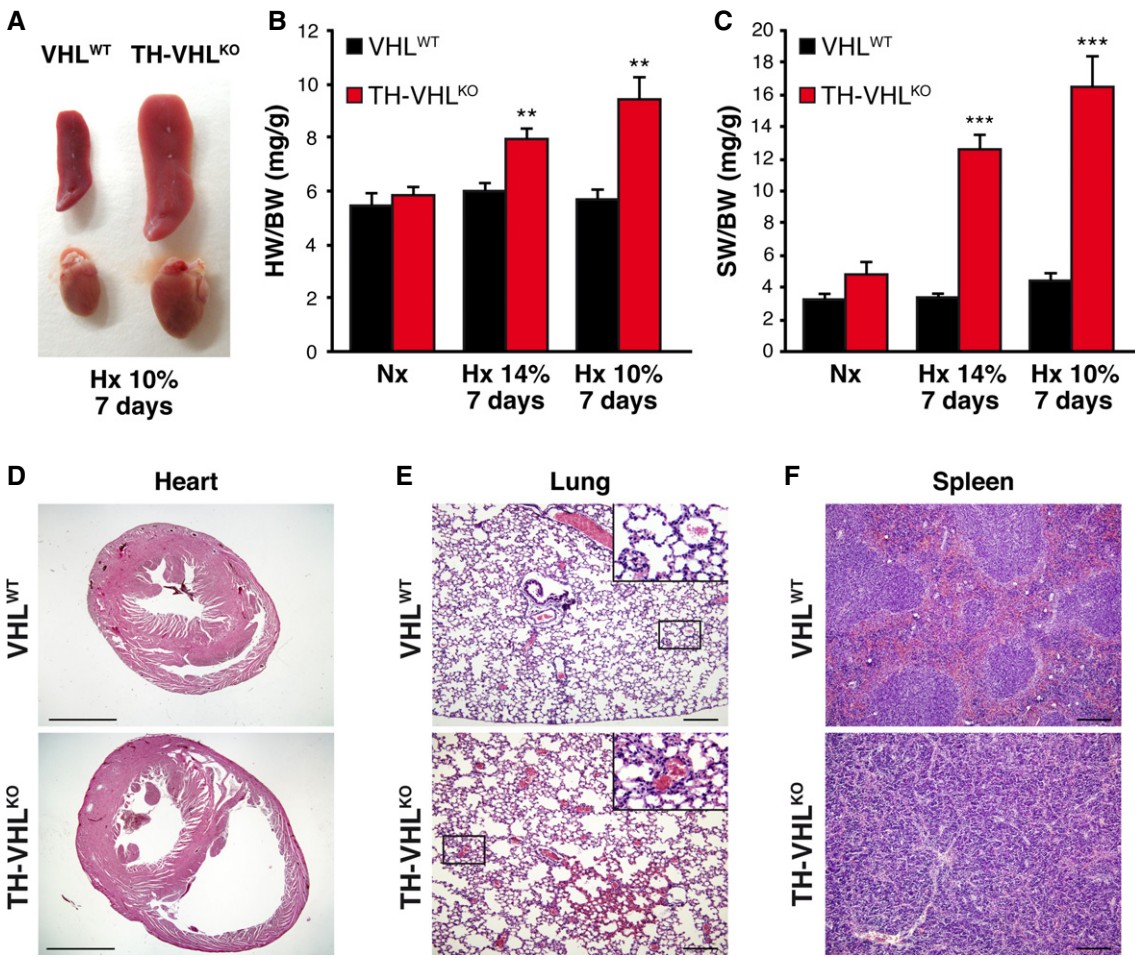

**Figure 9.   Histopathological alterations in TH-VHL$^{KO}$ mice exposed to chronic hypoxia.**

A    Enlargement of the spleen (top) and heart (bottom) of TH-VHL$^{KO}$ mice after 7 days' exposure to chronic hypoxia (10% O$_2$).

B, C   Average wet weight of hearts (B) and spleens (C) versus body weight in mice kept for 7 days at the indicated levels of hypoxia ($n$ = 6 per genotype and condition).
VHL$^{WT}$ versus TH-VHL$^{KO}$, **$P$ = 0.007 (14% O$_2$), **$P$ = 0.001 (10% O$_2$) in (B) and ***$P$ = 0.00006 (14% O$_2$), ***$P$ = 0.00003 (10% O$_2$) in (C) (unpaired two-tailed $t$-test).

D–F   Hematoxylin/eosin staining of thin sections of heart (D), lung (E), and spleen (F) of VHL$^{WT}$ and TH-VHL$^{KO}$ mice maintained for 7 days in chronic hypoxia. Note the marked increase in size of the right ventricle (D), as well as pulmonary edema (E) and changes in the histological architecture of the spleen (F). Scale bars: (D) 2 mm; (E, F) 200 μm. Nx, normoxia; Hx, hypoxia. All experiments were performed with 8- to 12-week-old mice. This figure is accompanied by Supplementary Fig S6.

sustentacular (GFAP$^+$) cells was impaired in the TH-VHL$^{KO}$ mice due to the loss of the floxed *Vhl* allele in TH$^+$ cells. Interestingly, the GFAP$^+$ progenitor cells in TH-VHL$^{KO}$ mice did not proliferate in response to sustained hypoxia. This observation fits well with the concept that neurotransmitter release from O$_2$-sensitive glomus cells (lost or profoundly impaired in the TH-VHL$^{KO}$ mice) is the signal that triggers the proliferation of progenitor cells to bring about CB growth during exposure to hypoxia (Platero-Luengo *et al*, 2014).

### Intolerance to hypoxia in mice with impairment of the acute O$_2$-sensing system

Acute HVR, a reflex response necessary for adaptation to hypoxic environments, disappears in patients that have undergone surgical CB resections (most commonly due to tumors or asthma treatment) (Timmers *et al*, 2003). These patients appear to live unaffected in normoxic environments, although disturbances during sleep and

cases of sudden death have been attributed to a lack of functional chemoreceptors (López-Barneo *et al*, 2008). Alterations of CB development have also been associated with respiratory dysfunction in neonates and children (for recent reviews see Perez & Keens, 2013; Porzionato *et al*, 2013; Gozal *et al*, 2013). TH-VHL$^{KO}$ mice, which also have a blunted acute HVR, seemed to live unaffected by this condition in normoxia, although their respiratory functions were not systematically analyzed here.

The CB is thought to play an essential role in acclimatization to chronic hypoxia, an environmental or medical condition affecting millions of people worldwide. Nevertheless, this process has been poorly investigated due to a lack of appropriate experimental models. The CB is generated during embryogenesis by the migration of sympathetic precursor cells from the SCG to the primordial carotid artery (see Hempleman & Warburton, 2013 for a detailed review). Mutations of genes that prevent either carotid artery formation or sympathetic development are known to result in CB defects.

However, these mutations are embryologically lethal or animals die shortly after birth due to major respiratory alterations, meaning that the animals cannot be studied in adulthood (Dauger *et al*, 2003; Kameda *et al*, 2008). The TH-VHL^KO is a novel animal model in which the consequences of functional inhibition of peripheral chemoreceptors can be studied throughout the normal life span of mice. When maintained under normoxic conditions, TH-VHL^KO mice show full development of the brain and other organs and normal physiological functions. Nonetheless, they exhibit a striking intolerance to sustained hypoxia. Even exposure of TH-VHL^KO mice to mild hypoxia (14% $O_2$), caused strong hemoglobin desaturation, which within a few days was followed by splenomegaly, severe pulmonary hypertension, and right cardiac hypertrophy leading to death. Therefore, these data demonstrate the absolute necessity of peripheral chemoreceptors for acclimatization and survival during exposure to hypoxia. These observations make TH-VHL^KO mice an ideal model to study the early signs of hypoxia intolerance or to identify biomarkers sensitive to maladaptation to hypoxia. This could help prevent hypoxia-associated morbidities affecting the brain or cardiorespiratory system, which are highly prevalent in susceptible individuals (Sutherland & Cherniack, 2004; Schou *et al*, 2012; Gozal *et al*, 2013).

## General pathophysiological consequences of sympathoadrenal atrophy

Besides a lack of functional peripheral chemoreceptors, the TH-VHL^KO mice had also atrophy of the peripheral sympathetic nervous system and decreased catecholamine (particularly adrenaline) secretion. These animals also showed hypoglycemia, which was particularly prominent in the fasting state, along with other signs of sympathetic dysfunction (data not shown). TH-VHL^KO mice could thus serve as an excellent model to test the role of the CB-AM axis in adaptation to situations involving an elevated $O_2$ demand (such as physical exercise) or the function of organs devoid of autonomic innervation such as, for example, the endocrine pancreas (Borden *et al*, 2013; Muñoz-Bravo *et al*, 2013) or bone marrow (Méndez-Ferrer *et al*, 2008). These animals could also help provide further insight into the role of the CB-AM axis in glucose homeostasis and blood pressure regulation. Experiments performed in animals (Koyama *et al*, 2000; Pardal & López-Barneo, 2002) and in man (Wehrwein *et al*, 2010; Ortega-Sáenz *et al*, 2013) have suggested that CB glomus cells are glucose sensors that participate in the acute counter-regulatory response to hypoglycemia; however, this function of the CB is under debate. On the other hand, it is well established that CB activation leads to increased sympathetic tone. CB inhibition produces marked effects on blood pressure in hypertensive rats (Paton *et al*, 2013), and CB denervation has been proposed as a therapeutic strategy to combat chronic neurogenic hypertension (McBryde *et al*, 2013).

# Materials and Methods

## Mice and animal care

Generation and genotyping of mouse strains carrying the *Vhl* conditional alleles, *Phd3* null alleles, and non-degradable variants of *Hif-1α* and *Hif-2α* conditional alleles has been described previously (Haase *et al*, 2001; Kim *et al*, 2006; Bishop *et al*, 2008). Catecholaminergic-specific *Vhl* ablation or *Hif-1α* and/or *Hif-2α* activation was achieved by mating these animals with Th-IRES-Cre and Th-IRES-CRE^ER transgenic mice (Lindeberg *et al*, 2004; Rotolo *et al*, 2008; Díaz-Castro *et al*, 2012). See online Supporting Information for further details on mouse lines. All procedures involving mice were performed in accordance with European Union guidelines (2010/63/EU) and Spanish law (R.D. 53/2013 BOE 34/11370-420, 2013) concerning the care and use of laboratory animals and were approved by the Ethics Committee of the University of Seville.

## Tissue preparation, immunohistochemistry, and electron microscopy

Mice were killed by an overdose of sodium pentobarbital injected intraperitoneally. The carotid bifurcation, adrenal gland, DRG, and abdominal sympathetic ganglia (celiac and mesenteric ganglia) were dissected, fixed for 2 h with 4% paraformaldehyde (Sigma), and cryopreserved (30% sucrose) for cryosectioning (10 μm thick). The distal part of the small intestine, as well as the brain, heart, lungs, and spleen, was removed, fixed overnight with 4% paraformaldehyde, embedded in paraffin, and sectioned at a thickness of 8 μm, with the exception of the brain which was sectioned at 20 μm. Further details of immunocytochemical procedures can be found in the online Supporting Information. Neuronal cell counting and area measurements were calculated from sections spaced 80 μm (DRG, myenteric ganglia, AM and SCG-CB TH$^+$ area) or 40 μm (CB) apart throughout the organ; this was performed on microscope images (Olympus BX61) using ImageJ software. The volume of the CB, AM, and CB-SCG area was estimated according to the Cavalieri's principle (Díaz-Castro *et al*, 2012). A blinded investigator performed all the histological quantifications. Hematoxylin and eosin staining was performed by following standard procedures. For electron microscope studies of CB and AM, we introduced some modifications in animal sacrifice, fixation, and tissue sectioning (see Supporting Information). All the images were taken using a transmission electron microscope (Philips CM-10) and a digital camera (Veleta).

## Generation and differentiation of carotid body neurospheres

Carotid body neurospheres were obtained and processed as indicated previously (Pardal *et al*, 2007). Details are provided in the online Supporting Information.

## Lentivirus production and gene silencing

Lentiviral vector (LV) production was carried out by transient cotransfection of either individual pGIPZ *Phd3, Hif-1α, Hif-2α* or pGIPZ non-silencing shRNA plasmid (Thermo Scientific) with the packaging plasmid pCMVΔR8.91 and the envelope plasmid pMD.G in 293T cells, according to previously published procedures (Macías *et al*, 2009). Lentiviral particle titers between $8 \times 10^5$ and $5 \times 10^6$ UT/ml were routinely obtained as determined by flow cytometric analysis of transduced green fluorescent protein (GFP)-positive 293T cells. The knockdown efficacy of each *Phd3, Hif-1,α or Hif-2α*

shRNA LV was validated by transducing a mouse embryonic fibroblast cell line (NIH3T3) at a multiplicity of infection (MOI) of 5 and subsequent use of qRT-PCR (see for details online Supporting Information). Dispersed CB cells were transduced (MOI = 10) with selected *Phd3, Hif-1,α or Hif-2α* shRNA LVs and then used for neurosphere formation and differentiation assays. For GFP detection on transduced flat neurosphere colonies, neurospheres were incubated with Alexa-Fluor 488-conjugated rabbit anti-GFP (1:500; Molecular Probes, A21311).

**Plethysmography and measurement of hemoglobin saturation**

Respiratory parameters were measured in conscious, unrestrained mice using whole-body plethysmography (Emka Technologies) according to the manufacturer's recommended configuration of the apparatus for use with mice. Animals were maintained in a hermetic chamber with controlled normoxic airflow until they were settled, following which they were exposed to hypoxic air (10% O$_2$) for 5 min, with normoxia again reinstated after this period. Each animal was subjected to this cycle of normoxia–hypoxia–normoxia twice per session. Control (VHL$^{WT}$ and VHL$^{WT}$;PHD3$^{KO}$) and mutant (TH-VHL$^{KO}$ and TH-VHL$^{KO}$;PHD3$^{KO}$) mice were routinely alternated between both chambers to avoid any intrinsic variability in the equipment. Real-time respiratory data were acquired and stored using iox2 software (Emka Technologies). Raw data obtained from plethysmography recordings were filtered by score rate, and only those that fully complied with the experimental protocol were analyzed. For the calculation of average respiratory parameters, we took into account values during the first 2 min once a 10% O$_2$ level had been reached inside the chamber, as determined by an oxygen probe (Greisinger Electronic). This corresponds to an O$_2$ level just above that before mutant TH-VHL$^{KO}$ and TH-VHL$^{KO}$;PHD3$^{KO}$ mice suffer respiratory depression or loss of consciousness. Arterial blood hemoglobin saturation was measured in conscious, slightly anesthetized unrestrained mice using MouseOx Plus (Starr Live Sciences Corp.) linked to the Emka plethysmography system to monitor real-time percent oxygen saturation while a stable normoxic or hypoxic (14% O$_2$ or 10% O$_2$) airflow was applied. Raw data were acquired and stored with iox2 software (Emka Technologies). Average percentages of hemoglobin saturation levels were determined throughout the first minute after a 14% O$_2$ or 10% O$_2$ had been reached.

**Chronic hypoxia and physiological parameters**

Mice (2–3 months old) were chronically exposed to a 14% O$_2$ or 10% O$_2$ environment by using a specially designed hermetic chamber with controlled O$_2$ and CO$_2$ levels and temperature and humidity monitoring (Coy Laboratory Products). After 7 days of exposure, animals were weighed, anesthetized with ketamine/xylazine (100 mg/kg body weight and 8 mg/kg body weight, respectively), and bled for subsequent hematocrit measurement and plasma collection. Plasma EPO levels were determined using the Quantikine Mouse EPO ELISA kit (R&D Systems) according to the manufacturer's protocol. Heart, lung, and spleen tissues were removed, and their wet weights measured. Next, tissues were histologically processed as described above. Analogous procedures were followed for mice maintained in normoxia.

**The paper explained**

**Problem**
Mutations of the *VHL* gene are associated with pheochromocytomas and paragangliomas; however, the pathogenesis of these disorders is unknown. Experiments on cell lines have suggested that VHL participates in the molecular cascade leading to the natural apoptosis of sympathetic progenitor cells. In this way, a deficit of this protein could predispose an organism to tumorigenesis. As the actual role of VHL on sympathoadrenal homeostasis is unknown, we sought to address this question by generating genetically modified mice lacking Vhl specifically in catecholaminergic cells.

**Results**
We show that *Vhl* inactivation does not lead to tumorigenesis but rather to a marked atrophy of the AM, CB, and sympathetic ganglia. Hypoxia-induced adult CB neurogenesis, an adaptive response characteristic of normal mammals, is also markedly inhibited in mice with the ablation of *Vhl* alleles. *Vhl*-deficient animals show signs of dysautonomia, but survive well under normoxic conditions. However, they exhibit a striking lack of acclimatization to hypoxia, which is characterized by erythrocytosis, pulmonary edema, and right cardiac hypertrophy leading to death.

**Impact**
Our findings indicate that, contrary to generally held beliefs ascribing a role to *VHL* as a tumor suppressor gene, *Vhl* inactivation in rodent catecholaminergic cells *in vivo* does not lead to tumorigenesis, but rather to a marked atrophy of the affected organs. Therefore, Vhl has differing functions in cells of diverse embryological origin or developmental stage. These observations explain why VHL disease characterized by the presence of pheochromocytomas is rarely associated with the bi-allelic loss of *VHL*. Vhl-deficient mice will thus serve as an unprecedented model to study the early signs of hypoxia intolerance, which in a translational medicine setting could help prevent hypoxia-associated morbidities in susceptible individuals.

**Statistical analysis**

Data are presented as the mean ± standard error of the mean (SEM). Statistical significance was assessed by the Student's *t*-test with a Levene test for determining the homogeneity of variances in cases of normal distribution, or by the nonparametric Mann–Whitney *U*-test in cases of non-normal distribution. Kaplan–Meier survival curves' statistical significance was analyzed by log-rank test. PASW18 software was used for all statistical analyses.

**Supplementary information** for this article is available online: http://embomolmed.embopress.org

**Acknowledgements**

This project was supported by grants from the Botín Foundation and the Spanish Ministry of Science and Innovation (SAF program). M. Carmen Fernández-Agüera received a predoctoral fellowship from the Spanish Ministry of Science and Innovation (FPI program). We thank Dr Peter J. Ratcliffe for having provided us with Phd3 knockout mice and Dr José A. Rodríguez-Gomez for help with blood catecholamine determinations. We also thank Dr Juan Luis Ribas (CITIUS, University of Seville) for help with electron microscopy and to personnel of the Core Facilities of IBiS for their technical support.

## Author contributions

DM and JL-B designed the experiments and wrote the manuscript; DM, MCF-A, and VB-H performed the experiments; and JL-B supervised the project.

## Conflict of interest

The authors declare that they have no conflict of interest.

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
