## [Review Process File · EMBO Molecular Medicine]

Deletion of the von Hippel-Lindau gene causes sympathoadrenal cell death and impairs chemoreceptor-mediated adaptation to hypoxia

David Macías, M. Carmen Fernández-Agüera, Victoria Bonilla-Henao and José López-Barneo

Corresponding author: José López-Barneo, University of Seville

Review timeline:

Submission date:	08 April 2014
Editorial Decision:	21 May 2014
Revision received:	05 September 2014
Editorial Decision:	09 October 2014
Revision received:	15 October 2014
Accepted:	20 October 2014

Transaction Report:

Editor: Céline Carret

1st Editorial Decision

21 May 2014

Thank you for the submission of your manuscript to EMBO Molecular Medicine. We have now heard back from the two referees whom we asked to evaluate your manuscript. In addition, we have consulted with an editorial advisor, which explains the delay in getting back to you.

Although the referees find the study to be of potential interest, you will see that they also raise a number of concerns mainly about the limited mechanistic insights and medical significance. As we feel that the suggested revisions are reasonable and would considerably improve the manuscript, we would encourage you to address all issues as best as possible. Please note that it is EMBO Molecular Medicine policy to allow only a single round of major revision and that, as acceptance or rejection of the manuscript will depend on another round of review, your responses should be as complete as possible.

I look forward to receiving your revised manuscript.

***** Reviewer's comments *****

Referee #1 (Comments on Novelty/Model System):

The TH-Cre;Vhl(f/f) mice are not a model of any known disease and do not help to explain any known physiological or pathological process.

Referee #1 (Remarks):

The authors analyze the effect of loss of VHL expression in cells expressing tyrosine hydroxylase. They find a loss of glomus cells in the carotid body that results in loss of the hypoxic ventilatory response. There are other responses to hypoxia that appear to occur in excess, but these much more interesting findings are only briefly mentioned in the final paragraph of the results section. The authors do not identify the molecular mechanism by which loss of VHL expression leads to loss of glomus cells. They perform studies to determine whether hypoxia-inducible factor 1alpha (HIF-1alpha) or prolyl hydroxylase domain protein 3 (PHD3) is involved, but do not test for involvement of HIF-2alpha, PHD1, or PHD2 by gene targeting studies. They state that the loss of glomus cells occurs by cell death but do not perform a single experiment that directly analyzes cell viability or mechanisms of cell death. It is not clear whether the data in Fig. 6E on "TH+ cell survival" can adequately distinguish between death of TH+ cells vs loss of TH expression.

Referee #2 (Remarks):

Macias, Lopez-Barneo and colleagues report that targeted genetic deletion of the von Hippel-Lindau (VHL) gene in murine catecholaminergic cells leads to defects in the development of the carotid body (CB), adrenal medulla and sympathetic ganglia. In particular the mice show deficiencies in tyrosine hydroxylase (TH)-positive type I glomus cells in the CB, while the number of type II (stem) cells is increased. This effect is attributed to decreased survival of the TH+ cells. Using a combination of cell culture assays and genetic mouse models the authors conclude that these effects are largely independent on the VHL targets HIF-1/2 and on the HIF-inducible prolyl hydroxylase 3 (PHD3) that has been linked to apoptosis; no alternative mechanisms have been examined. The CB defects lead to an inability of the mice to mount a normal physiological response to hypoxia, resulting in their eventual death under conditions where all control mice survive.

This manuscript deals with several topics of interest, including the role of VHL in CB development, cell survival and differentiation, the cell context-dependent distinction between tumor promoting and apoptosis promoting, as well as HIF-dependent and HIF-independent functions of VHL. The authors also propose that the TH-VHL-KO mice can be a suitable animal model for the study of the consequences of functional disruption of peripheral chemoreceptor cells in adaptation to hypoxia and possibly other metabolic stresses.

This study uses a number of genetic mouse models that are well suited for addressing the central questions of the manuscript in vivo. The experiments are well designed, performed and presented, and the data is generally of high quality.

I have the following remaining concerns:

1) The central phenotype reported in this study - reduction of the number of TH+ cells in the CB of VHL-KO mice in vivo (Fig. 1A, Fig. 6A) - has not been quantified. It is important to perform this quantification, as the claimed "striking decrease" and "strong reduction" in TH+ cell number is not apparent in all cases from the presented images.

2) Assuming that the loss of TH+ type I glomus cells is quantitatively confirmed, an important question raised by the manuscript is why the increased number of CB stem/progenitor (GFAP+) cells in TH-VHL-KO mice fails to give rise to a sufficient number of type I cells. The authors propose that while VHL KO progenitor cells can differentiate into TH+ cells, the latter die out. This model needs to be supported by examination and quantification of cell death, e.g. autophagy, as implied by the authors, or apoptosis, in which VHL has been shown to play a role. If increased

autophagy/apoptosis can be confirmed in the cells, the authors should examine whether autophagy/apoptosis inhibitors can rescue the loss of TH⁺ cells, at least in culture.

Technical points:

- 1) A Table S1 is cited on page 12, but I did not find it.
- 2) I see no reason why the graph in Fig. 3B should be flipped.
- 3) The legends on the graphs in Fig. 5C, 8C and particularly 8B, are not very easy to decipher - consider using colours instead of hatched patterns.
- 4.) Do the authors see increased HIF1/2 and PHD3 expression following loss of VHL? This would be nice to show to back up the validity of the rescue or phenocopy experiments in Fig.5/6/7.
- 5) I suppose that in Fig. 8B "TH-VHL-WT; PHD3-KO" should be "TH-VHL-KO; PHD3-KO", similarly in Fig. 8C "TH-VHL-WT" should be "TH-VHL-KO"
- 6) p. 9, line 11: should be Fig. 4E, not Fig 3E.

1st Revision - authors' response

05 September 2014

Referee #1 (Comments on Novelty/Model System):

The TH-Cre; Vhl(f/f) mice are not a model of any known disease and do not help to explain any known physiological or pathological process.

Authors' response: Respectfully, we disagree with this comment by the reviewer. Our TH-VHLKO mouse is a model of von Hippel-Lindau (VHL) disease restricted to catecholaminergic cells, which are those that undergo loss of heterozygosity in cases of pheochromocytoma or paraganglioma. As discussed in the paper, a non-explainable low occurrence of pheochromocytoma in individuals with bi-allelic loss of *VHL* (type I VHL disease) has been reported. The data in our paper not only provide an explanation for this observation, but also suggest that *VHL* mutations causing pheochromocytomas/paragangliomas are probably gain-of-function mutations (rather than loss of function as was previously thought).

Moreover, our TH-VHLKO animals, with atrophy of the carotid body-adrenal gland axis, are a model of intolerance to hypoxia, which is observed in some people suffering from acute/chronic mountain sickness or in individuals that have had the carotid bodies removed due to asthma or neck tumor surgery. Therefore, as shown in this paper, the TH-VHLKO mice provide an excellent rodent model to study the systemic effects of arterial chemoreceptor dysfunction. Arterial chemoreceptors are necessary for acclimatization to hypoxia, a condition that affects millions of people residing at high altitude or with restricted gas exchange in the lungs.

Finally, hyper-activation of the carotid body-adrenal medulla axis (as seen for example in sleep apnea patients or after cardiac failure) has been proposed to contribute to the autonomic imbalance involved in hypertension, insulin resistance and heart hypertrophy with poor prognosis). Indeed, carotid body resection is being clinically tested as a potential therapeutic option in chronic hypertension. Therefore, understanding the pathophysiology of the acute O₂-sensing system (particularly arterial chemoreceptors) is gaining increasing medical interest. As discussed in this paper, the TH-VHLKO mice serve as an excellent tool for such studies.

Referee #1 (Remarks):

The authors analyze the effect of loss of VHL expression in cells expressing tyrosine hydroxylase. They find a loss of glomus cells in the carotid body that results in loss of the hypoxic ventilatory response. There are other responses to hypoxia that appear to occur in excess, but these much more interesting findings are only briefly mentioned in the final paragraph of the results section.

Authors' response: We are not sure if we have understood this comment of the referee. The responses to hypoxia that occur in excess (erythrocytosis, cardiac hypertrophy, etc) are not due to primary loss of *Vhl*, as this gene is only deleted in catecholaminergic cells. They are secondary to the lack of hypoxic hyperventilatory response necessary for acclimatization to hypoxia. To clarify this concept we have added two sentences at the end of the "Results" section (page 13, lines 12 and 16-17).

The authors do not identify the molecular mechanism by which loss of *VHL* expression leads to loss of glomus cells.

Authors response: We would like to point out that our work is not done on a cell line, where there is a practically unlimited amount of tissue in which *Vhl* is deleted. On the contrary, it is done on small organs with relatively few catecholaminergic cells that are dying due to the lack of *Vhl*. The *in vitro* preparation of neurospheres from TH-VHLKO mice contains none or, rarely, one TH⁺ cell. Therefore it is practically impossible to do biochemical experiments on these preparations to investigate the causes of TH⁺ cell loss. Nonetheless, our electron microscope studies suggest that cell death is compatible with autophagy dysregulation; previous work done on other preparations, in which a large amount of tissue was available, has shown senescence, apoptosis and autophagy in *Vhl*-deficient cells (Young *et al.*, 2008; Mazure & Pouyssegur, 2010; Li & Kim, 2011). Following the comments of the reviewers we have investigated in neurospheres from TH-VHLKO animals whether the presence of drugs that inhibit autophagy, apoptosis or necroptosis have any effect on the survival of differentiated TH⁺ cells. These data are presented in Figure 1 below, which is included only for review purposes.

Figure 1. Carotid body neurospheres (8-days-old) generated from VHL^{WT} and THVHL^{KO} mice were cultured under differentiation conditions for 3 days with or without different cell death inhibitors. Pan caspase inhibitor Z-MAD-FMK (carbobenzoxy-valyl-alanyl-aspartyl-[O-methyl]-fluoromethylketone) 20 μ m. 3-MA (3-Methyladenine) 5mM. This drug inhibits autophagy by blocking autophagosome formation via the inhibition of type III Phosphatidylinositol 3-kinases (PI-3K). Nec-1 (Necrostatin-1) 30 μ m. This is an inhibitor of necroptosis (non-apoptotic cell death pathway). A cocktail of the three drugs was also used in some experiments. After 3 days of differentiation, cells were fixed and stained for TH by immunocytochemistry. The number of TH⁺ cells were counted and normalized respecting the number of neurospheres formed in each condition. N=4, p=0.043.

Only the cocktail of drugs had a small, but reproducible and statistically significant, effect on the survival of TH⁺ cells. Although in our view these data are not sufficiently robust to justify inclusion of the figure in the paper, a sentence describing these experiments has been added to the section where the mechanisms of cell death are discussed (page 15, last line and page 16, lines 1-5).

They perform studies to determine whether hypoxia-inducible factor 1alpha (HIF-1alpha) or prolyl hydroxylase domain protein 3 (PHD3) is involved, but do not test for involvement of HIF-2alpha, PHD1, or PHD2 by gene targeting studies.

Authors' response: Following this comment by the reviewer we have performed new experiments with lentiviral vectors to down-regulate *Hif-1alpha* and *Hif-2alpha* expression in carotid body neurospheres. As expected, none of these treatments had any effect on TH⁺ cell differentiation and survival. These new data are shown in the two plots of Fig. 5D and are also described in the text (page 10, lines 13-23).

Please note that we show in the paper data obtained from experiments with DMOG, a drug that inhibits all the prolyl hydroxylases (PHD1, PHD2 and PHD3) and stabilizes both Hif-1alpha and Hif-2alpha (Fig. 5B and C). We have also done experiments with transgenic animals over-

expressing Hif-1alpha, Hif-2alpha or both (Fig. 7).

They state that the loss of glomus cells occurs by cell death but do not perform a single experiment that directly analyzes cell viability or mechanisms of cell death.

Authors' response: We have tried hard to investigate the mechanisms of cell death. Indeed, these time- and resource-intensive experiments performed with double transgenic mice (TH-VHLKO plus PHD3KO or TH-VHLKO over-expressing Hif-1a, Hif-2a or both) were all designed to directly analyze the mechanism of cell death induced by Vhl deletion. However, as discussed above, our model, although useful to investigate the consequences of Vhl deficiency *in vivo*, is not appropriate to study in detail how the lack of Vhl can lead to catecholaminergic cell loss.

It is not clear whether the data in Fig. 6E on "TH+ cell survival" can adequately distinguish between death of TH+ cells vs loss of TH expression.

Authors' response: We agree with this comment by the referee as TH expression can be modulated by numerous factors and therefore the amount of active protein can change under different experimental conditions. However, we think that the information available suggests that Vhl deficiency leads to catecholaminergic cell death rather than to a decrease of TH expression for the following reasons.

1. The *in vivo* data demonstrates an obvious reduction in size of the superior cervical ganglion-carotid body (SCG-CB) area with alteration of the structure, which is not related to TH expression used as a marker. In addition, the complete loss of hypoxic ventilatory response, characteristic of these animals, cannot be explained by the down regulation of TH. It is well known that dopamine is not the transmitter that activates the afferent sensory fibers terminating at the respiratory center. An obvious decrease in the number of catecholaminergic cells (atrophy) is clear in the SCG-CB area and adrenal medulla of TH-VHLKO mice.
2. TH immunostaining of control and TH-VHLKO neurospheres was done in parallel. The intensity of TH fluorescence in individual cells was the same in the two types of neurospheres. It was the number of fluorescent (TH+) cells and not the intensity of the fluorescence what changed between wild type and Vhl deficient cells.
3. We have performed experiments looking at survival of TH+ cells at different times (3 and 10 days) in carotid body neurospheres. We show that Vhl deficiency does not interfere with the appearance of cells that are TH+ but rather affects the long-term survival of the cells (Fig. 6F, page 11, lines 13-18).

Referee #2 (Remarks):

Macias, Lopez-Barneo and colleagues report that targeted genetic deletion of the von Hippel-Lindau (VHL) gene in murine catecholaminergic cells leads to defects in the development of the carotid body (CB), adrenal medulla and sympathetic ganglia. In particular the mice show deficiencies in tyrosine hydroxylase (TH)-positive type I glomus cells in the CB, while the number of type II (stem) cells is increased. This effect is attributed to decreased survival of the TH+ cells. Using a combination of cell culture assays and genetic mouse models the authors conclude that these effects are largely independent on the VHL targets HIF-1/2 and on the HIF-inducible prolyl hydroxylase 3 (PHD3) that has been linked to apoptosis; no alternative mechanisms have been examined. The CB defects lead to an inability of the mice to mount a normal physiological response to hypoxia, resulting in their eventual death under conditions where all control mice survive. This manuscript deals with several topics of interest, including the role of VHL in CB development, cell survival and differentiation, the cell context-dependent distinction between tumor promoting and apoptosis promoting, as well as HIF-dependent and HIF-independent functions of VHL. The authors also propose that the TH-VHL-KO mice can be a suitable animal model for the study of the consequences of functional disruption of peripheral chemoreceptor cells in adaptation to hypoxia and possibly other metabolic stresses. This study uses a number of genetic mouse models that are well suited for addressing the central questions of the manuscript in vivo. The experiments are well designed, performed and presented, and the data is generally of high quality.

I have the following remaining concerns:

1) The central phenotype reported in this study - reduction of the number of TH+ cells in the CB of VHL-KO mice in vivo (Fig. 1A, Fig. 6A) - has not been quantified. It is important to perform this quantification, as the claimed "striking decrease" and "strong reduction" in TH+ cell number is not apparent in all cases from the presented images.

Authors' response: We agree with the reviewer that this point needed clarification. Following his/her comments we have quantified the volume occupied by TH+ cells at P0 and P60, as done in previous papers from our laboratory (Diaz-Castro et al., 2012; Platero-Luengo et al., 2014). These new data appear in Fig. 1B (page 6, lines 12-14) and Fig. 6B (page 11, lines 3-4). Plots in 1B and 6B clearly show that the extension of the TH+ parenchyma drastically decreases in animals with *Vhl* deficiency in catecholaminergic cells. We have quantified the superior cervical ganglion-carotid body (SCG-CB) area, since in the TH-VHLKO mice these two structures cannot be separated as the few existing CB cells are intermingled with SCG cells. The decrease in number of TH+ cells in the adrenal medulla is unequivocally observed in the histological photographs (see Fig. 1C panels at P60).

2) Assuming that the loss of TH+ type I glomus cells is quantitatively confirmed, an important question raised by the manuscript is why the increased number of CB stem/progenitor (GFAP+) cells in TH-VHL-KO mice fails to give rise to a sufficient number of type I cells. The authors propose that while VHL KO progenitor cells can differentiate into TH+ cells, the latter die out. This model needs to be supported by examination and quantification of cell death, e.g. autophagy, as implied by the authors, or apoptosis, in which VHL has been shown to play a role. If increased autophagy/apoptosis can be confirmed in the cells, the authors should examine whether autophagy/apoptosis inhibitors can rescue the loss of TH+ cells, at least in culture.

Authors' response: We agree with the reviewer on the importance of analyzing the mechanism of cell death induced by *Vhl* deficiency. However, we would like to point out that our work has not been done on a cell line, where there is a practically unlimited amount of tissue in which *Vhl* is deleted. On the contrary, the experiments have been performed on small organs with a small number of catecholaminergic cells that are dying due to the lack of *Vhl*. The time- and resource-intense experiments performed with double transgenic mice (TH-VHLKO plus PHD3KO or TH-VHLKO over-expressing Hif-1a, Hif-2a or both) were all designed to directly analyze the mechanism of cell death induced by *Vhl* deletion. They already indicate that PHD3 (a pro-apoptotic protein in the sympathoadrenal setting) may play a role, although ablation of *Phd3* does not inhibit the *Vhl* deficiency effect. On the other hand, the *in vitro* preparation of neurospheres from TH-VHLKO mice contains none or, rarely, one TH+ cell. Therefore it is practically impossible to do biochemical experiments on these preparations to investigate the causes of TH+ cell loss in TH-VHLKO mice. To comply with the referee requests we have done the following experiments:

1. Our electron microscope studies suggest that glomus or chromaffin cell death in THVHLKO mice is compatible with autophagy dysregulation; moreover previous work done on other preparations in which a large amount of tissue was available, has shown senescence, apoptosis and autophagy in *Vhl*-deficient cells (Young *et al.*, 2008; Mazure & Pouyssegur, 2010; Li & Kim, 2011). We have investigated in CB neurospheres from TH-VHLKO animals whether the presence of drugs that inhibit autophagy, apoptosis or necroptosis has any effect on the survival of differentiated TH+ cells. These data are presented in Figure 1 below, which is included only for review purposes.

Figure 1. Carotid body neurospheres (8-days-old) generated from VHL^{WT} and TH-VHL^{KO} mice were cultured under differentiation conditions for 3 days with or without different cell death inhibitors. Pan caspase inhibitor Z-MAD-FMK (carbobenzoxy-valyl-alanyl-aspartyl-[O-methyl]-fluoromethylketone) 20 μ m. 3-MA (3-Methyladenine) 5mM. This drug inhibits autophagy by blocking autophagosome formation via the inhibition of type III Phosphatidylinositol 3-kinases (PI-3K). Nec-1 (Necrostatin-1) 30 μ m. This is an inhibitor of necroptosis (non-apoptotic cell death pathway). A cocktail of the three drugs was also used in some experiments. After 3 days of differentiation, cells were fixed and stained for TH by immunocytochemistry. The number of TH+ cells were counted and normalized respecting the number of neurospheres formed in each condition. N=4, p=0.043.

Only the cocktail of drugs had a small, but reproducible and statistically significant, effect on the survival of TH+ cells. Although, in our view, these data are not sufficiently robust to justify inclusion of the figure in the paper, a sentence describing these experiments has been added to the section where the mechanisms of cell death are discussed (page 15, last line and page 16, lines 1-5).

2. We investigated apoptosis by TUNEL in cultured carotid body neurospheres and, rather surprisingly, found a large number of cells undergoing apoptosis in the growing colonies. This observation is presented in Figure 2 for review purposes.

Figure 2. Carotid body neurospheres (8-days-old) generated from VHL^{WT} and TH-VHL^{KO} mice were cultured under differentiation conditions for 3 days. Then, cells were fixed and stained for TUNEL (green fluorescence). The number of TH+ cells were counted and normalized respecting the number of neurospheres formed in each condition. Size bar: 200 μ m. Insets show in each case two cells marked with dapi (blue nuclei), one undergoing apoptosis.

This experiment does not help to study catecholaminergic cell death, as the number of TH+ cells in the neurospheres is very small (4-6 cells per neurosphere in VHL^{WT} and none or, rarely, one cell per neurosphere in TH-VHL^{KO}). TUNEL+ cells in these assays are most likely a pool of proliferating progenitors undergoing, at the same time, apoptosis. Although not related to the mechanism of catecholaminergic cell death, it is interesting that the number of apoptotic cells is

higher in wild type than in TH-VHLKO neurospheres. This observation fits quite well with the larger number and size of CB neurospheres obtained from TH-VHLKO mice in comparison with controls (page 8, last 10 lines and Fig. 4B and C).

3. We have performed experiments looking at survival of TH⁺ cells at different times (3 and 10 days) in carotid body neurospheres. We show that Vhl deficiency does not interfere with the appearance of cells that are TH⁺, but rather affects the long-term survival of the cells (Fig. 6F, page 11, lines 13-18).

Technical points:

1) *A Table S1 is cited on page 12, but I did not find it.*

Probably table S1 that was uploaded separately from other files, was not included in the pdf for review. In the revised documents table S1 is added to the Supplemental Information file.

2) *I see no reason why the graph in Fig. 3B should be flipped.*

The graph is now presented with a vertical orientation.

3) *The legends on the graphs in Fig. 5C, 8C and particularly 8B, are not very easy to decipher - consider using colours instead of hatched patterns.*

We have now included colors in the figures, as suggested by the referee.

4.) *Do the authors see increased HIF1/2 and PHD3 expression following loss of VHL? This would be nice to show to back up the validity of the rescue or phenocopy experiments in Fig.5/6/7.*

We have tried to do this but Hif-1alpha and Hif2-alpha antibodies did not work well in the SCG-CB tissue.

5) *I suppose that in Fig. 8B "TH-VHL-WT; PHD3-KO" should be "TH-VHL-KO; PHD3-KO", similarly in Fig. 8C "TH-VHL-WT" should be "TH-VHL-KO"*

The mistakes have been corrected

6) *p. 9, line 11: should be Fig. 4E, not Fig 3E.*

The error has been corrected

2nd Editorial Decision

09 October 2014

Thank you for the submission of your revised manuscript to EMBO Molecular Medicine. We have now received the enclosed reports from the referees who were asked to re-assess it.

As you will see, referee 1 is still not satisfied, but referee 2 (who had overlapping issues) is, pending minor changes. In this particular case, and as we do not feel that mechanistic insights are critical for publication of this article in EMBO Molecular Medicine, I am pleased to inform you that we will be able to accept your manuscript pending final amendments.

***** Reviewer's comments *****

Referee #1 (Remarks):

The authors show profound effects of VHL loss of function on the carotid body and adrenal medulla, which resulted in an impaired ventilatory response to hypoxia. Unfortunately, the authors were unable to establish a specific molecular target of VHL underlying the observed phenotype.

Referee #2 (Remarks):

The revised manuscript by Macias, Lopez-Barneo et al, together with the rebuttal letter, includes new experiments addressing points raised during the review process. Overall, the manuscript presents clear genetic evidence for an important function of VHL in the development of TH+ cells in the carotid body.

The new quantifications shown in Fig. 1B and 6B address my first concern about the extent of the principal phenotype. The text describes them as representing the „volume of the CB-SCG TH+ area", but the graphs and figure legends only say „CB-SCG volume". It should be made explicit whether the total volume or only the volume of TH+ cells was quantified - both are relevant and mentioned in the paper, but the latter is required to support the authors' basic claims.

The data in the manuscript and rebuttal letter do not provide clear evidence that cell death in TH+ cells is altered - and certainly not that this is the main factor causing the observed decrease in TH+ cell number. While the technical challenges that have hampered the resolution of this issue are appreciated, the authors should at least provide a more balanced discussion of this point, which for example also reflects the possibility of impaired TH cell differentiation or TH expression, as mentioned by reviewer 1.

2nd Revision - authors' response

15 October 2014

Please note that to comply with the comments of referee 2 we have made the following changes in the revised manuscript.

Reviewer: The new quantifications shown in Fig. 1B and 6B address my first concern about the extent of the principal phenotype. The text describes them as representing the "volume of the CB-SCG TH+ area", but the graphs and figure legends only say "CB-SCG volume". It should be made explicit whether the total volume or only the volume of TH+ cells was quantified - both are relevant and mentioned in the paper, but the latter is required to support the authors' basic claims.

Authors' response: Fig. 1B and Fig. 6B. The text in the ordinate axis have been modified to indicate that measurements refer to TH+ volume. Similar indications have been made in the figure legends.

Reviewer: The data in the manuscript and rebuttal letter do not provide clear evidence that cell death in TH+ cells is altered - and certainly not that this is the main factor causing the observed decrease in TH+ cell number. While the technical challenges that have hampered the resolution of this issue are appreciated, the authors should at least provide a more balanced discussion of this point, which for example also reflects the possibility of impaired TH cell differentiation or TH expression, as mentioned by reviewer 1.

Authors' response: We have modified the text in several places to make the point more open as indicated by the reviewer.

1. Page 9, lines 17-18. The text "...it appears that newly formed glomus cells die..." has been replaced with "... it appears that newly formed glomus cells are unable to differentiate to TH+ cells or die..."
2. Page 15, line 3 from the bottom and page 17, line 13. "...survival..." has been replaced with "...differentiation and/or survival..."
3. Page 16, line 7. "...required for maintenance..." has been replaced with "...required for maintenance (full differentiation and survival)..."

I hope that all the material necessary for the definitive acceptance and publication of the paper is now properly submitted. I appreciate very much your attention to our work and the fair treatment received during the reviewing process.